# TeamWork: Multivariate Time Series Anomaly Detection via Asymmetric Role-aware Channel Modeling

Shiyan Hu [1]   Tengxue Zhang [1]   Jianxin Jin [1]   Xiangfei Qiu [1]   Bin Yang [1]   Chenjuan Guo [1]

## Abstract

Multivariate time series anomaly detection remains challenging as it requires the joint modeling of variable relationships and temporal dependencies. Existing methods often struggle to balance channel relationship modeling and overlook the relative importance of different variables within multivariate time series. To address this, we propose TeamWork, an asymmetric role-aware channel modeling framework that decouples variables into dominant and auxiliary roles according to their contributions to uncertainty reduction. Dominant variables drive system evolution and their deviations more strongly disrupt normal patterns, while auxiliary variables provide complementary cues. These variables with different roles are integrated through a role-aware gated interaction module. Moreover, point and subsequence anomalies can exist in multiple periodic systems, and the same anomaly type may behave differently across short- and long-period series. To capture such variations, we introduce a period-aware masked modeling mechanism. It employs multiple specialized masking mechanisms spanning short to long periods to facilitate comprehensive temporal dependency learning. Extensive experiments on multiple real-world datasets demonstrate that TeamWork achieves superior performance compared with state-of-the-art methods.

## 1. Introduction

With the extensive deployment of multi-sensors in modern systems, multivariate time series anomaly detection (MT-SAD) has gained increasing importance. It aims to identify abnormal patterns from historical observations and has been successfully applied across various domains, such as indus-

trial system monitoring and financial fraud detection (Wen et al., 2022; Li et al., 2021; Yang et al., 2023a; Boniol et al., 2022; 2024; Sylligardos et al., 2023). More broadly, time series modeling has also played an important role in forecasting, irregular-series analysis, and robust decision support across real-world scenarios (Qiu et al., 2025b; 2026a; Liu et al., 2026b;a; Wu et al., 2025b;a; Yu et al., 2025b;a; Wang et al., 2026a;b; Wu et al., 2026). However, since MTSAD involves simultaneously modeling complex temporal dynamics and inter-variable (a.k.a., channel) [1] dependencies, it remains a highly challenging task.

From the perspective of channel modeling, existing MT-SAD methods can be broadly categorized into channel-independent (CI) strategy and channel-dependent (CD) strategy. CI strategy (Figure 1a) treats each channel independently (Nie et al., 2022; Li et al., 2025; Shentu et al., 2025), failing to capture inter-channel dependencies in scenarios where natural physical correlations exist among variables. In contrast, CD strategy (Figure 1b) indiscriminately models dependencies across all channels (Wang et al., 2023), which often leads to redundant interactions and increases the risk of overfitting (Han et al., 2024). Therefore, recent studies have introduced channel-clustering (CC) strategy (Wu et al., 2025c; Qiu et al., 2025c), which simplifies dependency modeling by clustering channels with similar patterns and isolating the rest. As shown in Figure 1c, the CC strategy applies CD modeling within the cluster and CI modeling among dissimilar channels. It explicitly captures inter-channel dependencies while mitigating overfitting in CD modeling, leading to improved anomaly detection performance.

Despite their success, the CC strategy still fundamentally overlooks the relative importance of different variables within multivariate time series. According to the maximum entropy principle, the evolution of a complex system can be explained by a small subset of dominant and informative variables (Shannon, 2001; Zeng et al., 2025). In anomaly detection, anomalies occurring in such dominant variables are more likely to distort normal temporal patterns and thus deserve greater attention. Meanwhile, the auxiliary variables, though less influential in driving system evolution, still carry complementary information correlated with the

---

[1]East China Normal University, Shanghai, China. Correspondence to: Chenjuan Guo <cjguo@dase.ecnu.edu.cn>.

*Proceedings of the 43rd International Conference on Machine Learning*, Seoul, South Korea. PMLR 306, 2026. Copyright 2026 by the author(s).

---

[1]We use the terms "variable" and "channel" interchangeably.

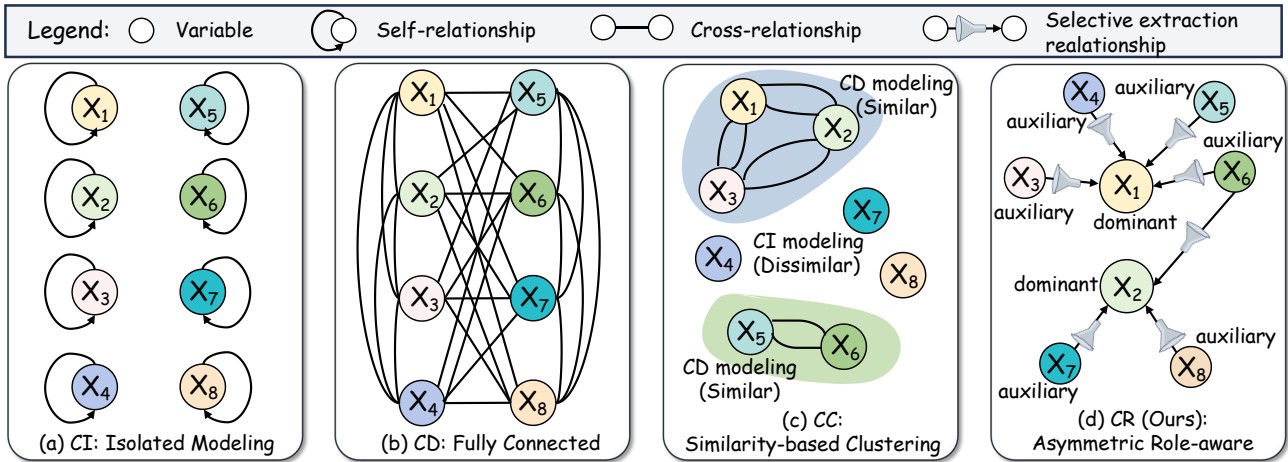

*Figure 1.* Schematic comparison of channel modeling strategies: (a) CI, (b) CD, (c) CC, and (d) our proposed CR.

dominant ones and should not be disregarded. However, CC clusters channels solely based on their pairwise similarity, rather than identifying the most representative variables that truly drive system dynamics. As a result, intra-cluster channels tend to exhibit highly similar temporal patterns, leading to homogeneous information redundancy. Furthermore, unclustered channels may still contain heterogeneous yet valuable information that remains underutilized by CC, as it treats these channels independently.

Moreover, existing methods often overlook the fact that the same anomaly type may behave differently across short- and long-period series. Anomalies generally manifest as either point anomalies or subsequence anomalies (Lai et al., 2021; Qiu et al., 2025a). Point anomalies tend to be more prominent in short-period series, but may be smoothed out in long-period series and thus become harder to identify. In contrast, subsequence anomalies are more evident in long-period series but can be difficult to detect effectively in short-period series if their duration exceeds the period length. In practice, real-world time series often exhibit multi-periodic patterns (Wu et al., 2023), such as daily and weekly variations in traffic flow, monthly and annual variations in e-commerce transactions. However, common point-wise and patch-wise modeling methods (Yang et al., 2023b; Wang et al., 2023; Shentu et al., 2025; Li et al., 2025; Wu et al., 2025c; Campos et al., 2021) are largely agnostic to such multi-periodicity, which limits their ability to capture the diverse anomaly behavior in multi-periodic systems.

To address these challenges, we propose **TeamWork**, a multivariate time series anomaly detection via asymmetric role-aware channel modeling.

For the first challenge, we propose an innovative channel role-aware (CR) mechanism, grounded in the maximum entropy principle, to explicitly assign channels to asymmetric roles (see Figure 1d). Specifically, we quantify the importance of each variable by measuring the entropy reduction

it induces in the system. Variables that contribute most significantly to uncertainty reduction are designated as dominant, while the remaining ones are classified as auxiliary. Furthermore, to effectively leverage the complementary information provided by auxiliary variables without introducing redundancy, we design a role-aware gated interaction module. This module uses the learned representations of dominant variables to steer the model in extracting informative and relevant information from auxiliary variables.

For the second challenge, we propose a novel period-aware masked modeling mechanism to capture the diverse manifestations of anomalies in multi-periodic systems. Specifically, taking the dominant variables as input, we construct multi-period series matrices that explicitly encode temporal dynamics at different periodic resolutions. These matrices are subsequently processed by period-aware maskers tailored to different anomaly characteristics. For point anomaly detection, short-period series matrices are equipped with a point masker to capture fine-grained deviations. For subsequence anomaly detection, middle-period series matrices are combined with a segment masker to capture context-aware patterns, and long-period series matrices are paired with a period masker to capture cross-period dependencies.

Our contributions can be summarized as follows:

- We propose an asymmetric role-aware channel modeling mechanism that distinguishes dominant variables driving system evolution from auxiliary variables, then fuses the complementary information of auxiliary variables through a role-aware gated interaction module.

- We propose a period-aware masked modeling mechanism to accommodate the diverse manifestations of anomalies in multi-periodic systems. A Point masker, a segment masker, and a period masker are applied to series matrices of different periods, respectively.

- Extensive experiments on multiple real-world benchmark datasets demonstrate that TeamWork consistently outperforms state-of-the-art MTSAD methods, validating the effectiveness of role-aware variable modeling and period-aware temporal dependency learning.

## 2. Related Work

### 2.1. Multivariate Time Series Anomaly Detection

In recent years, deep learning methods have demonstrated superior performance and attracted increasing attention. Recent studies further extend time series analysis to anomaly prediction, multimodal anomaly detection, streaming anomaly detection, dataset condensation, and cross-modality modeling (Hu et al., 2024; 2026a;b; Miao et al., 2025; 2024; Liu et al., 2025). Traditional MTSAD methods can be categorized into non-learning (Breunig et al., 2000; Goldstein & Dengel, 2012; Yeh et al., 2016) and machine learning (Liu et al., 2008; Ramaswamy et al., 2000; Shyu et al., 2003; Yairi et al., 2001). They can be roughly classified into three main categories: forecasting-based methods (Deng & Hooi, 2021), contrastive-based methods (Yang et al., 2023b; Xu et al., 2021), and reconstruction-based methods (Shentu et al., 2025; Wang et al., 2023; Dai et al., 2024; Wu et al., 2025c; Li et al., 2025; Luo & Wang, 2024; Campos et al., 2021).

Forecasting-based methods predict current or future values from historical observations and treat the prediction error as the anomaly criterion. Contrastive-based methods identify anomalies by measuring discrepancies between normal and abnormal samples in the embedding space. Reconstruction-based methods encode the input time series into a compressed representation and then decode it to reconstruct the original sequence, where reconstruction errors serve as anomaly scores. Mask reconstruction, as an effective self-supervised learning paradigm, is generally classified into point-wise methods (Xu et al., 2021; Wang et al., 2023; Song et al., 2023) and patch-wise methods (Yang et al., 2023b; Li et al., 2025; Shentu et al., 2025). Point-wise methods apply masking at individual time steps to learn fine-grained temporal representations. In contrast, patch-wise methods divide the input series into multiple patches, applying masking and encoding at the patch level to capture temporal patterns within and across patches. Although these methods have achieved impressive performance in MTSAD, most of them rely on fixed, period-agnostic masking strategies. Such designs fail to adequately account for the periodic variations inherent in time series, which limits their effectiveness in real-world scenarios.

### 2.2. Channel Modeling in MTSAD

In MTSAD, channel modeling falls into three main strategies: channel-independent (CI) strategy (Nie et al., 2022; Zeng et al., 2023; Yang et al., 2023b), channel-dependent (CD) strategy (Liu et al., 2023; Zhao et al., 2020), and channel-clustering (CC) strategy (Wu et al., 2025c; Qiu et al., 2025c). Related correlation-aware ideas have also been explored in forecasting with exogenous variables and time-series foundation/adaptation models (Qiu et al., 2026b; Li et al., 2026c; Cheng et al., 2026b;a; Lu et al., 2026a;b; Shang et al., 2026; 2024).

CI strategy, such as PatchTST (Nie et al., 2022), focuses solely on modeling temporal dependencies within individual channels. As a result, they fail to capture intrinsic inter-channel dependencies in scenarios where natural physical correlations exist among channels. Existing research has indicated that correlations within channels are crucial for effective time series anomaly detection (Song et al., 2018). Therefore, the CD strategy, exemplified by iTransformer (Liu et al., 2023), explicitly models dependencies across all channels. However, indiscriminately modeling all channel interactions may introduce irrelevant or spurious dependencies, thereby increasing the risk of overfitting. As a result, a recent study CATCH (Wu et al., 2025c), explores a CC strategy that simplifies dependency modeling by grouping channels with similar patterns while isolating the remaining ones. However, they are prone to learning redundant patterns within clusters and struggle to effectively exploit information from unclustered channels. Furthermore, existing CC strategies ignore identifying the most representative variables that truly govern system dynamics. More references for analysis can be found in Appendix C.

## 3. Preliminaries

### 3.1. Problem Definition

In multivariate time series anomaly detection, we consider an input time series of length $T$ as $\mathbf{X} = \{x_{i,j}\}_{i=1,j=1}^{D,T} \in \mathbb{R}^{D \times T}$, where $D$ is the number of variables and $x_{i,j}$ is the observation of the $i$-th variable at timestamp $j$. The $x_{i,:} \in \mathbb{R}^T, i \in \mathcal{D}$ corresponds to the complete time series of the $i$-th variable. The goal of MTSAD is to determine whether the observation $x_{:,t} \in \mathbb{R}^D$ at time $t$ is anomalous.

### 3.2. Dominant and Auxiliary Variable Definition

Since different variables contribute unequally to the dynamics of multivariate time series systems, we introduce the concepts of dominant and auxiliary variables to capture the most informative components. Inspired by the maximum entropy principle (Shannon, 2001; Zeng et al., 2025), which suggests that variations in multivariate entropy can be used to characterize the evolution of the explanatory power of individual variables, we propose the following definition:

**Definition 3.1** (Dominant variables). For any variable $x_{i,:} \in \mathbf{X}$, if the knowledge of $x_{i,:}$ leads to a greater reduction in the uncertainty of the overall multivariate system than that induced by the remaining variable set $\mathbf{X} \backslash \{x_{i,:}\}$, then the variable $x_{i,:}$ is considered to be more representative,

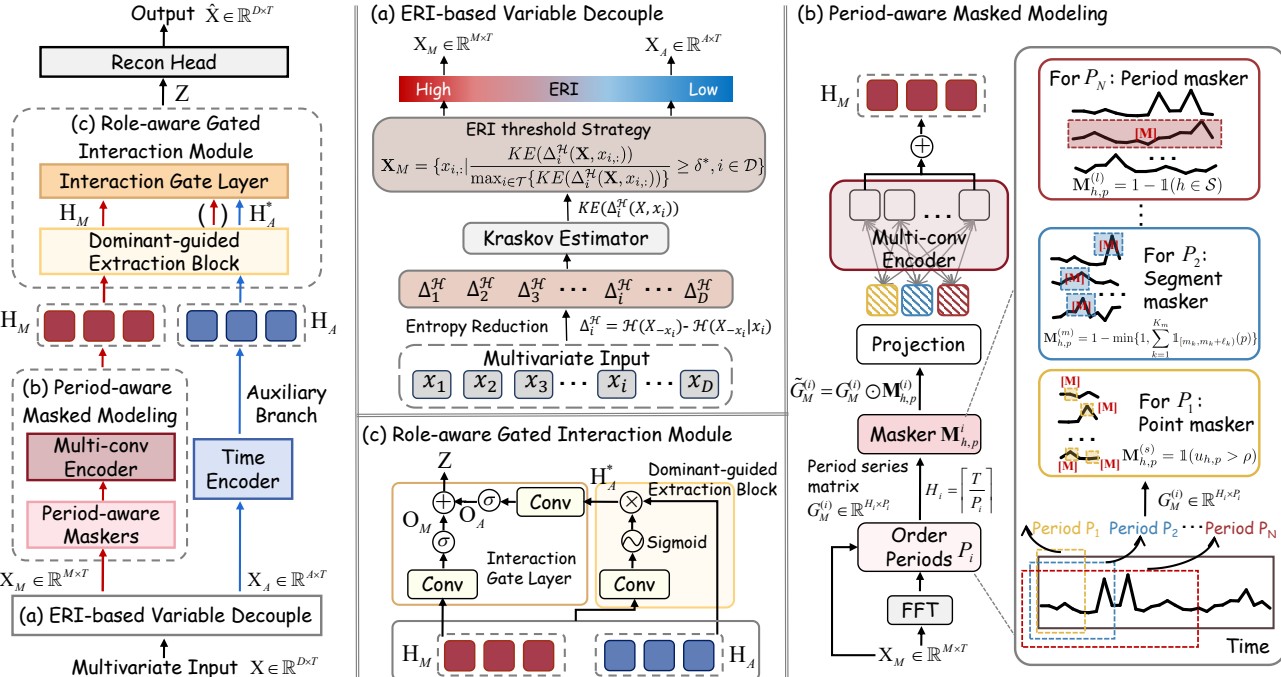

*Figure 2.* The overall structure of TeamWork, which consists of: (a) ERI-based Variable Decouple is devised to decouple dominant and auxiliary variables from the input $\mathbf{X}$; (b) Period-aware Masked Modeling enhances temporal dependency learning by assigning distinct masking strategies to series matrices of different periods; (c) Role-aware Gated Interaction Module integrates variables of different roles.

where $\mathbf{X}\backslash\{x_{i,:}\}$ denotes the variable set expect for $x_{i,:}$ in $\mathbf{X}$. Variables that induce the largest reductions in this conditional uncertainty are referred to as dominant variables.

Based on this definition, auxiliary variables are naturally characterized as those that carry comparatively less information about the overall multivariate time series, i.e., variables other than the dominant ones. Therefore, it is crucial to design an effective strategy to decouple multivariate and to model dominant and auxiliary variables accordingly.

## 4. Methodology

As illustrated in Figure 2, **TeamWork** advances multivariate time series anomaly detection through an asymmetric role-aware channel modeling framework that explicitly accounts for the heterogeneous roles of variables in dynamical complex systems. The framework is composed of three tightly coupled components: **(i) ERI-based Variable Decouple** (see Section 4.1), which decouples the input multivariate time series $\mathbf{X} \in \mathbb{R}^{D \times T}$ into variables that play dominant roles in characterizing the overall system behavior and those that assume auxiliary roles and provide complementary information; **(ii) Period-aware Masked Modeling** (see Section 4.2), which divides the time series of dominant variables $\mathbf{X}_M \in \mathbb{R}^{M \times T}$ into multi-period series matrices $\mathbf{G}_M^{(i)}$, and then applies period-aware maskers to generate the corresponding masked matrices $\tilde{\mathbf{G}}_M^{(i)}$; **(iii) Role-aware Gated Interaction Module** (see Section 4.3), which integrates rep-

resentations $\mathbf{Z}$ of variables with different roles through a role-aware gated interaction module that selectively incorporates complementary information from auxiliary variables. Notably, auxiliary variables $\mathbf{X}_A \in \mathbb{R}^{A \times T}$ are not modeled with the same level of complexity as dominant variables. Instead, a lightweight multilayer perceptron (MLP) is employed to derive auxiliary variable representations $\mathbf{H}_A$. This asymmetric design not only improves computational efficiency but also reduces the risk of introducing irrelevant or redundant information that may arise from overly complex modeling of auxiliary variables. Finally, a reconstruction head produces the model output $\hat{\mathbf{X}}$, and the reconstruction error is used as the anomaly score. The detailed design of each module is presented in the following subsections.

### 4.1. ERI-based Variable Decouple

To quantify the extent to which a single variable reduces the uncertainty of the entire system, we introduce the metric of entropy reduction index (ERI). The information entropy of the $i$-th variable $x_{i,:}$ is denoted as $\mathcal{H}(x_{i,:}) = -\sum_{\xi_i^t \in x_{i,:}} p(x_{i,:}) \log p(x_{i,:})$, where $\xi_i^t$ is the element at time $t$ of variable $x_{i,:}$. Similarly, the information entropy of variable set excluding $x_{i,:}$ is denoted as $\mathcal{H}(\mathbf{X}_{-x_{i,:}})$, where $\mathbf{X}_{-x_{i,:}} = \mathbf{X}\backslash\{x_{i,:}\}$. Then, the conditional entropy $\mathcal{H}(\mathbf{X}_{-x_{i,:}}|x_{i,:})$ is introduced to characterize the remaining uncertainty of the system given the knowledge of $x_i$. Hence,

the ERI $\Delta_i^{\mathcal{H}}(\mathbf{X}, x_{i,:})$ of variable $x_{i,:}$ is denoted as:

$$\Delta_i^{\mathcal{H}}(\mathbf{X}, x_{i,:}) = \mathcal{H}(\mathbf{X}_{-x_{i,:}}) - \mathcal{H}(\mathbf{X}_{-x_{i,:}}|x_{i,:}), \quad (1)$$

Furthermore, we adopt a relative ratio normalization strategy, in which each variable is evaluated with respect to the variable exhibiting the maximum ERI. This design preserves the intrinsic ranking of variables while providing a scale-invariant measure of importance. Specifically, the normalized ERI score $\delta_i$ of $i$-th each variable is defined as:

$$\delta_i = \frac{\Delta_i^{\mathcal{H}}(\mathbf{X}, x_{i,:})}{\bar{\Delta}^{\mathcal{H}}}, \quad (2)$$

where $\bar{\Delta}^{\mathcal{H}} = \max_{i \in \mathcal{D}}\{\Delta_i^{\mathcal{H}}(\mathbf{X}, x_{i,:})\}$ denotes the maximum ERI among all variables, and $\delta_i \in [0, 1]$. A threshold $\delta^*$ is then introduced to identify the set of dominant variables $\mathbf{X}_M$ from the multivariate time series as follows:

$$\mathbf{X}_M = \{x_{i,:}|\delta_i \geq \delta^*, i \in \mathcal{D}\}. \quad (3)$$

The threshold is applied to the normalized ERI rather than the absolute entropy-reduction value, because raw ERI magnitudes can vary across datasets with different scales and distributions. This normalization is order-preserving, i.e., it maintains the ranking of variables according to their relative explanatory power, while reducing sensitivity to dataset-specific entropy scales.

The definition of entropy, we prove that there exists an equivalent relationship between the ERI and mutual information (see details in Appendix B.1). This equivalence allows us to compute the ERI efficiently via mutual information in the subsequent analysis. The exact computation of $\Delta_i^{\mathcal{H}}(\mathbf{X}, x_{i,:})$ is intractable in practice. To address this, we employ the Kraskov estimator (KE) based on the k-nearest neighbor (Kraskov et al., 2004) to obtain an approximation, namely $\Delta_i^{\mathcal{H}}(\mathbf{X}, x_{i,:}) \approx KE(\Delta_i^{\mathcal{H}}(\mathbf{X}, x_{i,:}))$, as detailed in Appendix B.2. Accordingly, equation (3) is rewritten as:

$$\mathbf{X}_M = \{x_{i,:}|\frac{KE(\Delta_i^{\mathcal{H}}(\mathbf{X}, x_{i,:}))}{\max_{i \in \mathcal{D}}\{KE(\Delta_i^{\mathcal{H}}(\mathbf{X}, x_{i,:}))\}} \geq \delta^*, i \in \mathcal{D}\}. \quad (4)$$

Through this procedure, the set of dominant variables $\mathbf{X}_M$ is separated from $\mathbf{X}$, and the remaining variables constitute the set of auxiliary variables, denoted by $\mathbf{X}_A = \mathbf{X}\backslash\mathbf{X}_M$.

### 4.2. Period-aware Masked Modeling

As shown in Figure 2, period-aware masked modeling consists of two components: 1) *period-aware maskers* and 2) *a multi-conv encoder*. The period-aware maskers assign specific masking strategies to series matrices associated with different periods, while the multi-conv encoder then processes the masked series matrix using convolutional kernels with multiple receptive fields.

**Period-aware Maskers.** To account for the fact that the same anomaly type may manifest differently in short- and long-period series, we design a masker library and assign a tailored masking strategy to each period-specific series matrix. The period extraction is performed on dominant variables. Specifically, we first apply the Fast Fourier Transform (FFT) (Brigham & Morrow, 1967) to $\mathbf{X}_M$ and identify the top-$N$ dominant periods in ascending order, denoted as $\{P_1, P_2, \ldots, P_N\}$. In practice, we set $N = 3$ to preserve the most informative periodic components, thereby reducing the risk of relying on a single imperfect period estimate.

For each period $P_i$, $\mathbf{X}_M$ is reshaped into a period series matrix $\mathbf{G}_M^{(i)} \in \mathbb{R}^{H_i \times P_i}$, where $H_i = \left\lceil \frac{T}{P_i} \right\rceil$ denotes the number of periods. Each row corresponds to observations within a single period, while each column represents temporally aligned positions across different periods. To effectively capture the diverse periodic structures exhibited in time series, we design three complementary masking strategies tailored to period series matrices, namely the *Point Masker*, the *Segment Masker*, and the *Period Masker* as follows:

- **Point Masker.** For short-period series matrix $\mathbf{G}_M^{(s)}$, we employ a point-wise masking strategy that removes individual observations to capture fine-grained temporal dependencies. To construct the binary mask matrix $\mathbf{M}_{h,p}^{(s)}$, we utilize an independent stochastic sampling approach with a uniform prior. Specifically, for each entry indexed by $(h, p)$, we introduce an random variable $u_{h,p}, \sim \mathcal{U}(0, 1), \forall h \in \{1, \ldots, H_s\}, p \in \{1, \ldots, P_s\}$. The mask is defined by an indicator function $\mathbb{1}(\cdot)$ as:

$$\mathbf{M}_{h,p}^{(s)} = \mathbb{1}(u_{h,p} > \rho), \quad (5)$$

where $\rho \in (0, 1)$ represents the masking ratio. As a result, the masked input $\tilde{\mathbf{G}}_M^{(s)}$ is computed through the Hadamard product $\tilde{\mathbf{G}}_M^{(s)} = \mathbf{G}_M^{(s)} \odot \mathbf{M}_{h,p}^{(s)}$.

- **Segment Masker.** For middle-period series matrix $\mathbf{G}_M^{(m)}$, we apply a segment-wise masking strategy that removes contiguous temporal segments within each period to capture context-aware patterns. Specifically, we first define a sequence of masking intervals within period $h$ denoted as $\{(m_k, \ell_k)\}_{k=1}^{K_m}$, where the starting index $m_k$ is sampled uniformly from the valid range, i.e., $m_k \sim \mathcal{U}\{0, \ldots, P_m - \ell_k\}$. The segment length is $\ell_k = \lfloor \alpha P_m \rfloor$, with $\alpha \in (0, 1)$ controlling the relative length of each masked segment. The mask entry at temporal index $p$ is then computed as:

$$\mathbf{M}_{h,p}^{(m)} = 1 - \min\{1, \sum_{k=1}^{K_m} \mathbb{1}_{[m_k, m_k+\ell_k)}(p)\}, \quad (6)$$

where the indicator function $\mathbb{1}_{[m_k, m_k+\ell_k)}(p)$ equals 1

if $p \in [m_k, m_k + \ell_k)$ and 0 otherwise. Similarly, we obtain the masked input as: $\tilde{\mathbf{G}}_M^{(m)} = \mathbf{G}_M^{(m)} \odot \mathbf{M}_{h,p}^{(m)}$.

- **Period Masker**: For the long-period series matrix $\mathbf{G}_M^{(l)}$, we employ a period-wise masking strategy that operates along the period dimension, masking all temporal indices $p$ within the selected period to effectively capture cross-period dependencies. We define the mask as follows:

$$\mathbf{M}_{h,p}^{(l)} = 1 - \mathbb{1}(h \in \mathcal{S}), \tag{7}$$

where $\mathcal{S}$ denotes the set of masked period indices, which are uniformly sampled according to a predefined masking ratio. Similarly, we obtain the masked input as: $\tilde{\mathbf{G}}_M^{(l)} = \mathbf{G}_M^{(l)} \odot \mathbf{M}_{h,p}^{(l)}$.

Overall, the point masker focuses on fine-grained temporal modeling and is well-suited for identifying point anomalies, where individual time points exhibit significant deviations from normal patterns. In contrast, the segment and period maskers encourage the model to capture broader contextual and cross-period temporal dependencies. Therefore, these coarser-grained masking strategies are more effective at revealing subsequence anomalies, where a continuous segment of data collectively deviates from typical behavior.

**Multi-conv Encoder.** Given the period series matrices with different masking patterns produced by the *Period-aware Masker*, we employ a multi-convolution encoder to model temporal patterns at multiple receptive fields. The encoded representation $E_M^{(i)}$ for the $i$-th period is computed as:

$$\mathbf{E}_M^{(i)} = \sum_{r=1}^{R} \text{Conv}_r \left( \text{Projection}(\tilde{\mathbf{G}}_M^{(i)}) \right), \tag{8}$$

where $R$ is the number of convolution kernels with different receptive fields (i.e., $1 \times 1$, $3 \times 3$, and $5 \times 5$). After encoding, an *Align* operation is applied to reshape each period representation along the temporal dimension and truncate padded positions, thereby recovering a unified temporal length $T$. Finally, representations from different periods are aggregated to obtain the final representation of the dominant variables $\mathbf{H}_M$:

$$\mathbf{H}_M = \text{Agg} \left( \left\{ \text{Align} \left( \mathbf{E}_M^{(i)}; P_i \right) \right\}_{i \in \{s,m,l\}} \right). \tag{9}$$

### 4.3. Role-aware Gated Interaction Module

The role-aware gated interaction module consists of two components: 1) the *dominant-guided extraction block*, and 2) the *interaction gate layer*.

**Dominant-guided Extraction Block.** Considering that auxiliary variables may convey information complementary to

that of the dominant variables, we introduce a supplementary feature extraction block that exploits the representations of dominant variables to guide the learning of auxiliary features. Specifically, a guidance signal is generated by jointly encoding the dominant and auxiliary representations, which is then employed to adaptively modulate the auxiliary representations $\mathbf{H}_A^*$:

$$\mathbf{H}_A^* = \mathbf{H}_A \odot \sigma \left( \text{Conv}_{1 \times 1}([\mathbf{H}_M; \mathbf{H}_A]) \right), \tag{10}$$

where $[\cdot; \cdot]$ denotes concatenation operation and $\sigma(\cdot)$ denotes the sigmoid activation function.

**Interaction Gate Layer.** Finally, the representations corresponding to the two roles are adaptively regulated through a gating mechanism to control the information flow, thereby mitigating redundancy and facilitating stable integration:

$$\mathbf{O}_M = \sigma \left( \text{Conv}_{1 \times 1}(\mathbf{H}_M) \right) \odot \text{Conv}_{1 \times 1}(\mathbf{H}_M), \tag{11}$$

$$\mathbf{O}_A = \sigma \left( \text{Conv}_{1 \times 1}(\mathbf{H}_A^*) \right) \odot \text{Conv}_{1 \times 1}(\mathbf{H}_A^*). \tag{12}$$

The gated outputs are subsequently fused to form the integrated representation $\mathbf{Z} = \mathbf{O}_M + \mathbf{O}_A$.

### 4.4. Anomaly Criterion

After the role-aware gated interaction module produces the integrated representation $\mathbf{Z}$ from variables with different roles, an MLP-based projection head is employed to reconstruct the original multivariate time series:

$$\hat{\mathbf{X}} = \text{Projection}(\mathbf{Z}) \in \mathbb{R}^{T \times C}, \tag{13}$$

where $\hat{\mathbf{X}}$ denotes the reconstructed time series.

The model is trained by minimizing the reconstruction loss, which encourages faithful recovery of normal temporal patterns. Specifically, the squared Frobenius norm is adopted as the reconstruction objective:

$$\mathcal{L}_{\text{rec}} = \left\| \mathbf{X} - \hat{\mathbf{X}} \right\|_F^2. \tag{14}$$

During the inference, the reconstruction error serves as the anomaly score, where larger reconstruction errors indicate a higher likelihood of anomalies.

## 5. Experiments

### 5.1. Experimental Settings

**Datasets.** We evaluate the performance of Teamwork on 8 real-world multivariate time series datasets: GECCO, Genesis, CalIt2, Creditcard, MSL, SMAP, PSM, and SMD. Further details about these datasets are provided in Appendix A.1, and the complete experimental results can be found in the Appendix D.

*Table 1.* Comparison results on multiple multivariate real-world datasets. The Aff-F and A-R are the Affiliated-F1 and AUC-ROC. The best results are highlighted in bold, and the second-best results are underlined.

| Method | GECCO | | Genesis | | CalIt2 | | Credit | | MSL | | SMAP | | PSM | | SMD | |
|---|---|---|---|---|---|---|---|---|---|---|---|---|---|---|---|---|
| Metric | Aff-F | A-R | Aff-F | A-R | Aff-F | A-R | Aff-F | A-R | Aff-F | A-R | Aff-F | A-R | Aff-F | A-R | Aff-F | A-R |
| HBOS | 0.708 | 0.557 | 0.721 | 0.897 | 0.756 | 0.798 | 0.695 | 0.951 | 0.680 | 0.574 | 0.509 | 0.585 | 0.658 | 0.620 | 0.629 | 0.626 |
| PCA | 0.785 | 0.711 | 0.814 | 0.815 | 0.768 | 0.790 | 0.710 | 0.871 | 0.678 | 0.552 | 0.505 | 0.396 | 0.702 | 0.648 | 0.738 | 0.679 |
| IF | 0.424 | 0.619 | 0.788 | 0.549 | 0.402 | 0.775 | 0.634 | 0.860 | 0.584 | 0.524 | 0.512 | 0.487 | 0.620 | 0.542 | 0.626 | 0.664 |
| Ocsvm | 0.666 | 0.804 | 0.677 | 0.733 | 0.783 | 0.804 | 0.714 | 0.953 | 0.641 | 0.524 | 0.503 | 0.393 | 0.531 | 0.619 | 0.742 | 0.602 |
| AE | 0.823 | 0.769 | 0.854 | 0.931 | 0.587 | 0.767 | 0.561 | 0.909 | 0.625 | 0.562 | 0.430 | 0.395 | 0.707 | 0.650 | 0.439 | 0.774 |
| NLinear | 0.882 | 0.936 | 0.829 | 0.755 | 0.757 | 0.695 | 0.742 | 0.948 | 0.723 | 0.592 | 0.610 | 0.437 | 0.843 | 0.585 | 0.844 | 0.738 |
| DLinear | 0.893 | 0.947 | 0.856 | 0.696 | 0.793 | 0.752 | 0.738 | 0.954 | 0.725 | 0.624 | 0.607 | 0.391 | 0.831 | 0.580 | 0.841 | 0.728 |
| Patch | 0.906 | 0.949 | 0.856 | 0.685 | 0.793 | 0.808 | 0.746 | 0.957 | 0.724 | 0.637 | 0.629 | 0.441 | 0.831 | 0.586 | 0.845 | 0.736 |
| TsNet | 0.894 | 0.954 | 0.864 | 0.913 | 0.794 | 0.771 | 0.744 | 0.957 | 0.734 | 0.613 | 0.637 | 0.442 | 0.842 | 0.592 | 0.831 | 0.727 |
| DC | 0.687 | 0.555 | 0.776 | 0.659 | 0.697 | 0.527 | 0.632 | 0.504 | 0.694 | 0.507 | 0.681 | 0.500 | 0.682 | 0.501 | 0.675 | 0.502 |
| ATrans | 0.782 | 0.516 | 0.856 | 0.947 | 0.729 | 0.533 | 0.650 | 0.552 | 0.692 | 0.508 | 0.692 | 0.501 | 0.710 | 0.514 | 0.724 | 0.508 |
| DualTF | 0.701 | 0.714 | 0.810 | 0.937 | 0.751 | 0.574 | 0.663 | 0.703 | 0.588 | 0.576 | 0.674 | 0.465 | 0.725 | 0.600 | 0.679 | 0.631 |
| iTrans | 0.839 | 0.795 | 0.891 | 0.690 | 0.812 | 0.791 | 0.713 | 0.934 | 0.710 | 0.611 | 0.577 | 0.409 | 0.854 | 0.592 | 0.827 | 0.745 |
| Modern | 0.893 | 0.952 | 0.833 | 0.676 | 0.780 | 0.676 | 0.744 | 0.957 | 0.726 | 0.633 | 0.616 | 0.434 | 0.825 | 0.593 | 0.840 | 0.722 |
| CATCH | 0.908 | 0.970 | 0.896 | 0.974 | 0.835 | 0.838 | 0.750 | 0.958 | 0.740 | 0.664 | 0.578 | 0.431 | **0.859** | 0.652 | 0.847 | 0.811 |
| CrossAD | 0.894 | 0.985 | 0.852 | 0.812 | 0.786 | 0.786 | 0.732 | 0.948 | 0.704 | 0.594 | 0.582 | 0.440 | 0.843 | 0.615 | 0.852 | 0.748 |
| **TeamWork** | **0.934** | **0.992** | **0.928** | **0.994** | **0.852** | **0.850** | **0.756** | **0.963** | **0.744** | **0.681** | **0.864** | 0.592 | 0.857 | **0.720** | **0.854** | **0.825** |

*Table 2.* Multi-metrics results on three real-world multivariate datasets. The best ones are in bold.

| Datasets | Metric | Acc | P | R | F1 | R-P | R-R | R-F | Aff-P | Aff-R | Aff-F | A-R | A-P | R-A-R | R-A-P | V-ROC | V-PR |
|---|---|---|---|---|---|---|---|---|---|---|---|---|---|---|---|---|---|
| GECCO | ModernTCN | 0.984 | 0.373 | 0.779 | 0.504 | 0.086 | 0.644 | 0.152 | 0.808 | **0.998** | 0.893 | 0.952 | 0.447 | 0.978 | 0.459 | 0.975 | 0.461 |
| | CATCH | 0.984 | 0.380 | **0.818** | 0.518 | 0.065 | 0.795 | 0.119 | 0.832 | **0.998** | 0.908 | 0.970 | 0.418 | 0.990 | 0.473 | 0.987 | 0.465 |
| | CrossAD | 0.983 | 0.362 | 0.785 | 0.496 | 0.074 | **0.804** | 0.136 | 0.810 | 0.997 | 0.894 | 0.985 | **0.494** | 0.990 | 0.497 | 0.990 | 0.491 |
| | TeamWork (ours) | **0.989** | **0.495** | 0.626 | **0.553** | **0.312** | 0.635 | **0.419** | **0.883** | 0.992 | **0.934** | **0.992** | 0.440 | **0.993** | **0.537** | **0.993** | **0.531** |
| Genesis | ModernTCN | 0.965 | 0.015 | 0.120 | 0.027 | 0.014 | 0.174 | 0.026 | 0.728 | **0.974** | 0.833 | 0.676 | 0.015 | 0.981 | 0.015 | 0.729 | 0.015 |
| | CATCH | 0.992 | 0.116 | 0.160 | 0.134 | 0.119 | **0.497** | 0.192 | 0.835 | 0.966 | 0.896 | 0.974 | 0.249 | 0.727 | 0.384 | 0.978 | 0.371 |
| | CrossAD | 0.976 | 0.030 | 0.160 | 0.051 | 0.042 | 0.223 | 0.070 | 0.762 | 0.966 | 0.852 | 0.812 | 0.020 | 0.867 | 0.025 | 0.859 | 0.024 |
| | TeamWork (ours) | **0.996** | **0.429** | **0.180** | **0.254** | **0.333** | 0.233 | **0.274** | **0.896** | 0.962 | **0.928** | **0.994** | **0.420** | **0.995** | **0.596** | **0.995** | **0.581** |
| SMD | ModernTCN | 0.931 | 0.151 | 0.145 | 0.148 | 0.092 | 0.378 | 0.148 | 0.755 | 0.948 | 0.840 | 0.722 | 0.130 | 0.743 | 0.130 | 0.742 | 0.130 |
| | CATCH | 0.918 | 0.194 | 0.305 | 0.237 | 0.095 | 0.478 | 0.158 | **0.773** | 0.938 | 0.847 | 0.811 | 0.172 | 0.800 | 0.159 | 0.797 | 0.159 |
| | CrossAD | 0.865 | 0.134 | **0.410** | 0.202 | 0.059 | **0.625** | 0.107 | 0.754 | **0.979** | 0.852 | 0.748 | 0.149 | 0.768 | 0.145 | 0.765 | 0.144 |
| | TeamWork (ours) | **0.919** | **0.196** | 0.307 | **0.239** | **0.109** | 0.514 | **0.180** | **0.773** | 0.952 | **0.854** | **0.825** | **0.215** | **0.831** | **0.219** | **0.830** | **0.219** |

**Baselines.** We conduct a comprehensive comparison of our model against 15 baselines, including the latest state-of-the-art (SOTA) anomaly detection models: PCA (Shyu et al., 2003), IForest (IF) (Liu et al., 2008), HBOS (Goldstein & Dengel, 2012), One-Class SVM (Ocsvm) (Schölkopf et al., 1999), ModernTCN (Modern) (Luo & Wang, 2024), TimesNet (TsNet) (Wu et al., 2023), DCdetector (DC) (Yang et al., 2023b), Anomaly Transformer (ATrans) (Xu et al., 2021), PatchTST (Patch) (Nie et al., 2022), iTransformer (iTrans) (Liu et al., 2023), AutoEncoder (AE) (Sakurada & Yairi, 2014), DLinear (Zeng et al., 2023), NLinear (Zeng et al., 2023), DualTF (Nam et al., 2024), CATCH (Wu et al., 2025c), and CrossAD (Li et al., 2025). Among these baselines, CrossAD and PatchTST are representative SOTA under CI strategies; iTransformer and TimesNet are SOTA among CD strategies; and CATCH is the current SOTA based on the CC strategy. Additional details of the baseline methods are provided in Appendix A.2.

**Setup.** Following established conventions in consistency with prior studies, we employ two primary evaluation metrics: the label-based Affiliated-F1 score (Aff-F) (Huet et al., 2022) and the score-based Area under the Receiver Operating Characteristics Curve (A-R) (Fawcett, 2006). For label-based evaluation, the anomaly decision threshold is selected from the predefined threshold-ratio set $[0.1, 0.5, 1, 2, 3, 5, 10, 15, 20, 25]\%$, ensuring a consistent evaluation setting across methods. Full results across a total of 16 evaluation metrics are reported in the Appendix A.3. More implementation details are presented in the Appendix A.4.

## 5.2. Detection Results

To ensure a fair comparison, we re-evaluat all baselines under the unified benchmark (Qiu et al., 2025a). Therefore, some deviations are expected, and we believe such a unified evaluation is more objective and reliable than directly comparing results from different papers under different settings. We evaluate TeamWork against 16 competitive baselines on 8 real-world datasets, as summarized in Table 1. Experimental results show that our method achieves superior performance across all datasets under both the Aff-F and the A-R metrics, demonstrating its strong capability to balance false-positive and true-positive rates across various pre-selected thresholds. Furthermore, as shown in Table 2, TeamWork also ranks among the top-performing methods across a wide range of additional evaluation metrics, reinforcing its effectiveness in accurately distinguishing between normal and abnormal samples. More comprehensive results, including

Table 3. Ablation study on different design choices.

| Variations | | Credit | Genesis | SMD | Avg |
|---|---|---|---|---|---|
| **Channel Variations** | CI | 0.913 | 0.886 | 0.792 | 0.864 |
| | CD | 0.948 | 0.938 | 0.812 | 0.899 |
| | CC | 0.921 | 0.885 | 0.809 | 0.872 |
| | w/o $\mathbf{X}_M$ | 0.552 | 0.727 | 0.740 | 0.673 |
| | w/o $\mathbf{X}_A$ | 0.948 | 0.957 | 0.802 | 0.902 |
| | random assignment | 0.905 | 0.886 | 0.811 | 0.867 |
| **Interaction Variations** | w/o extraction | 0.909 | 0.992 | 0.823 | 0.908 |
| | w/o gate layer | 0.908 | 0.985 | 0.820 | 0.904 |
| | add technique | 0.905 | 0.958 | 0.804 | 0.889 |
| | $\mathbf{X}_A$ influences $\mathbf{X}_M$ | 0.947 | 0.993 | 0.812 | 0.917 |
| **$\mathbf{X}_A$ modeled same as $\mathbf{X}_M$** | | 0.951 | 0.942 | 0.821 | 0.905 |
| **Masking Variations** | point masker | 0.943 | 0.978 | 0.812 | 0.911 |
| | segment masker | 0.954 | 0.981 | 0.820 | 0.918 |
| | period masker | 0.942 | 0.983 | 0.819 | 0.915 |
| | w/o masker | 0.916 | 0.963 | 0.791 | 0.890 |
| **TeamWork (ours)** | | **0.963** | **0.994** | **0.825** | **0.927** |

detailed metric evaluation, are provided in Appendix D.

## 5.3. Model Analysis

We analyze the effectiveness of ERI-based variable decouple, period-aware masked modeling, role-aware gated interaction module, and visualize the anomaly scores.

**Ablation study.** To assess the impact of different modules in TeamWork, we conduct ablation studies on: (1) Channel variations: replace the proposed asymmetric role-aware channel modeling strategy with existing channel modeling mechanisms, further evaluate variants that utilize only a subset of role-specific variables, and replace ERI-based decoupling with random role assignment. (2) Interaction variations: remove components of the role-aware gated interaction module, and consider a variant where the interaction module is replaced with a simple add operation, and introduce bidirectional guidance where $\mathbf{H}_A$ also modulates $\mathbf{H}_M$. (3) Auxiliary variables modeling: the auxiliary variables are modeled identically to the period-aware masked modeling used for dominant variables. (4) Masking variations: period-aware masked modeling is replaced with fixed masking strategies, and all maskers are removed.

The results, summarized in Table 3, yield the following observations: 1) Our proposed CR strategy outperforms all existing channel modeling approaches, further validating the effectiveness of the dominant-auxiliary variable discovery mechanism. Moreover, removing either dominant or auxiliary variables leads to performance degradation, underscoring the complementary roles that both types of variables play in anomaly detection. In contrast, random role assignment also results in a noticeable drop, since it fails to reliably identify variables that are most representative of the system. 2) Removal of any component of the interaction module harms performance. The most severe drop occurs when the two role-specific representations are fused via a naive add operation, highlighting the necessity of the role-aware gated interaction module for effectively integrating information across variable roles. Moreover, allowing $\mathbf{H}_A$ to further guide $\mathbf{H}_M$ does not yield clear improvement. A possible reason is that additional guidance from auxiliary variables may introduce redundant or noisy information into the dominant branch. 3) Applying the same period-aware masked modeling to auxiliary variables as used for dominant variables does not yield performance gains. This is likely because auxiliary variables contain relatively limited information, and overly fine-grained modeling may instead introduce noise and hinder learning. 4) Replacing the proposed period-aware masking strategy with fixed masking schemes degrades model performance. Fixed masking is insufficient to adapt to the diverse manifestations of anomalies in multi-periodic time series. In addition, removing the masker library makes the self-supervised reconstruction task overly simple, thereby weakening the model's ability to learn scale-specific anomaly representations. These results collectively demonstrate the necessity of assigning different masking strategies according to the underlying periodic characteristics. More ablation experiments can be found in Appendix E.2.

**Parameter Sensitivity.** We further investigate the sensitivity of TeamWork to key hyperparameters. Figure 4 shows the performance under different dominant-variable selection thresholds $\delta^*$ and time series mask ratios $m$. For $\delta^*$, changing the threshold only adjusts the strength of the dominant–auxiliary role partition, while all variables remain involved in anomaly detection through either the dominant branch or the auxiliary branch. Figure 4 (a) shows that TeamWork maintains stable performance over a broad range of $\delta^*$ values. Similarly, Figure 4(b) shows that the model is relatively stable across a wide range of mask ratios. TeamWork uses three specialized maskers to capture point-level, segment-level, and period-level anomaly patterns at different temporal scales. Thus, as long as the masking ratio remains within a reasonable range, the model can still learn effective representations from the multi-period masking structure. We find that setting both hyperparameters to approximately 0.5 generally leads to better performance. Therefore, we fix $\delta^* = 0.5$ and $m = 0.5$ as practical default settings in all experiments. Additional analytical experiments are presented in Appendix E.1.

**Anomaly criterion visualization.** To illustrate how TeamWork identifies anomalies, we visualize its behavior on different anomaly types in Figure 3. The first row shows the original time series with ground-truth anomaly locations, while the second row displays the corresponding anomaly scores produced by TeamWork. For point anomalies (first and second columns) that manifest as isolated abnormal points, the anomaly scores exhibit sharp spikes at true anomaly locations. In contrast, subsequence anomalies (third, fourth, and fifth columns) manifest as sustained

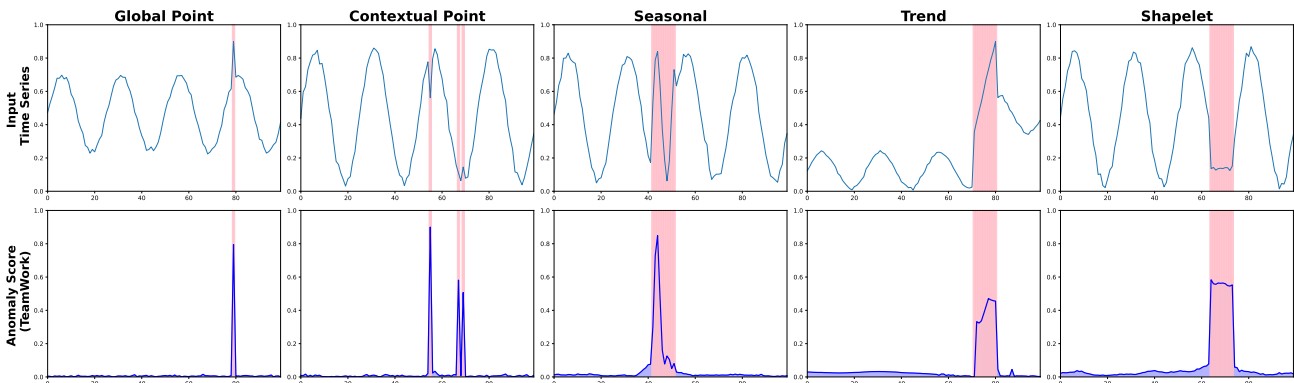

*Figure 3.* Visualization of anomaly scores from TeamWork across five anomaly patterns. The first row shows the original series. The second row displays anomaly scores based on the TeamWork.

disruptions over consecutive time intervals, breaking the underlying periodic structure. These results show that Team-Work can robustly detect different types of anomalies. More detailed visualization cases can be found in Appendix F.

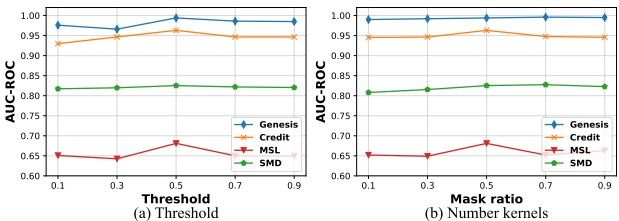

*Figure 4.* Results of the sensitivity analysis. Higher AUC-ROC (vertical axis) indicates better performance.

**Limitations.** Despite the advancements achieved by Team-Work, several limitations remain. First, the current framework adopts a static role perception mechanism, where dominant and auxiliary variables are determined according to their overall explanatory power. In highly dynamic systems, however, a variable that is dominant during one operating stage may become auxiliary in another stage. Second, the period-aware masked modeling module mainly focuses on global periodic structures extracted by FFT. When a time series exhibits weak periodicity or highly localized non-stationary temporal variations, the current global-period design may become less effective. Future work will investigate dynamic role-aware mechanisms that allow variable roles to evolve over time, and will combine TeamWork with local time-frequency tools, such as wavelet transforms, to better capture non-stationary temporal patterns.

## 6. Conclusion

In this work, we propose TeamWork, a novel framework for multivariate time series anomaly detection based on asymmetric role-aware channel modeling. TeamWork explicitly differentiates dominant variables that govern system evolution from auxiliary variables that provide complementary information, and integrates an entropy-based variable selec-

tion strategy with a role-aware gated interaction module to decouple and fuse role-specific representations. To capture diverse anomaly patterns in multi-period series, we further introduce a period-aware masked modeling mechanism that employs three specialized masking strategies for short- and long-periods. Experiments on multiple real-world benchmarks show that TeamWork outperforms SOTA methods, offering new insights into role-aware variable relationship modeling in multivariate time series.

## Impact Statement

Time series anomaly detection is of critical importance across a wide range of real-world applications, including finance and industry. In this study, all datasets used are publicly available, ensuring transparency and reproducibility of the experimental results. The contributions of this research advance the state of the art in time series anomaly detection. Our work may have various societal implications, but none require specific emphasis in this context.

## Acknowledgments

This work was partially supported by National Natural Science Foundation of China (62372179).

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

# A. Experimental details

## A.1. Datasets

The Table 4 provides a summary of the statistics for the publicly available real-world multivariate time series datasets (Lai et al., 2021; Hundman et al., 2018; Su et al., 2019; Qiu et al., 2025a), including GECCO, Genesis, CalIt2, Creditcard, MSL, SMAP, PSM, and SMD. The anomaly ratio varies from 0.17% to 11.07%, the range of feature dimensions varies from 2 to 55, and the sequence length varies from 5,040 to 1,416,825. These datasets are selected for their rich variation in data characteristics and anomaly types, which enables a thorough and systematic evaluation of multivariate time series anomaly detection methods.

*Table 4.* Statistics and descriptions of datasets used for multivariate time series anomaly detection. AR (%) denotes anomaly ratio.

| Dataset | Dim | AR (%) | Avg Total Length | Avg Test Length | Domain | Description |
|---|---|---|---|---|---|---|
| GECCO | 9 | 1.25 | 138,521 | 69,261 | Water treatment | Water quality dataset released in the GECCO Industrial Challenge. |
| Genesis | 18 | 0.31 | 16,220 | 12,616 | Machinery | A collection of sensor measurements and control signals obtained from cyber-physical manufacturing systems. |
| CalIt2 | 2 | 4.09 | 5,040 | 2,520 | Visitors flowrate | Pedestrian flow dataset that tracks the number of people entering and exiting a building over 15 days. |
| Creditcard | 29 | 0.17 | 284,807 | 142,404 | Finance | Financial transaction dataset containing credit card operations recorded over two days. |
| MSL | 55 | 5.88 | 132,046 | 73,729 | Spacecraft | Mars science laboratory dataset, provided by NASA, consists of telemetry records capturing the operational conditions of onboard sensors and actuators of the Mars rover. |
| SMAP | 25 | 9.72 | 562,800 | 427,617 | Spacecraft | Soil moisture active passive dataset, released by NASA, contains soil moisture measurements collected through satellite-based monitoring systems. |
| PSM | 25 | 11.07 | 220,322 | 87,841 | Server Machine | Pooled server metrics dataset is collected from eBay's server infrastructure and records various indicators related to system performance. |
| SMD | 38 | 2.08 | 1,416,825 | 708,420 | Server Machine | Server machine dataset records resource usage metrics collected from computer clusters operated by a large Internet company. |

## A.2. Baselines

We extensively compare TeamWork against 16 baselines, including: PCA (Shyu et al., 2003), IForest (IF) (Liu et al., 2008), HBOS (Goldstein & Dengel, 2012), One-Class SVM (Ocsvm) (Schölkopf et al., 1999), ModernTCN (Modern) (Luo & Wang, 2024), TimesNet (TsNet) (Wu et al., 2023), DCdetector (DC) (Yang et al., 2023b), Anomaly Transformer (ATrans) (Xu et al., 2021), PatchTST (Patch) (Nie et al., 2022), iTransformer (iTrans) (Liu et al., 2023), AutoEncoder (AE) (Sakurada & Yairi, 2014), DLinear (Zeng et al., 2023), NLinear (Zeng et al., 2023), DualTF (Nam et al., 2024), CATCH (Wu et al., 2025c), and CrossAD (Li et al., 2025).

- CrossAD (Li et al., 2025): A cross-scale reconstruction model that associates among multiple scales by reconstructing fine-grained series from coarser series.

- CATCH (Wu et al., 2025c): Enhances subsequence anomaly detection through frequency-domain patching and integrates fine-grained adaptive channel correlations across frequency bands.

- NLinear (Zeng et al., 2023): Introduce a set of embarrassingly simple one-layer linear models that surprisingly outperform existing sophisticated Transformer-based models.

- DLinear (Zeng et al., 2023): Divide the time series into the trend series and the remainder series, and then model these two series using two single-layer linear networks.

- AutoEncoder (AE) (Sakurada & Yairi, 2014): Autoencoders with nonlinear dimensionality reduction can detect subtle anomalies that linear PCA fails to capture, and their detection accuracy can be further improved by extending them to denoising autoencoders.

- ModernTCN (Modern) (Luo & Wang, 2024): Adopts a purely convolutional architecture to decouple and model temporal, channel-wise, and variable-wise relationships in multivariate time series.

- TimesNet (TsNet) (Wu et al., 2023): Employs a modular structure to decompose complex temporal patterns into different frequency components and maps one-dimensional time series into a two-dimensional space to model intra- and inter-period dynamics jointly.

- DCdetector (DC) (Yang et al., 2023b): Uses contrastive learning from both patch-wise and point-wise perspectives to discriminate between normal and anomalous patterns.

- Anomaly Transformer (ATrans) (Xu et al., 2021): Based on the hypothesis that anomalies exhibit stronger associations with nearby time points, it uses a minimax strategy to amplify association differences and enhance anomaly discrimination.

- PatchTST (Patch) (Nie et al., 2022): Applies channel-independent patching to multivariate time series, improving the model's ability to capture localized temporal features.

- DualTF (Nam et al., 2024): Employs a dual-domain architecture with nested sliding windows, where outer and inner windows handle time and frequency domains, respectively, aligning their anomaly scores to enhance detection.

- iTransformer (iTrans) (Liu et al., 2023): Embeds time information into variable tokens and applies attention mechanisms to model multivariate correlations.

- PCA (Shyu et al., 2003): Detects anomalies by measuring the deviation of samples in the principal component space, assuming anomalies lie far from the normal data distribution.

- One-Class SVM (Ocsvm) (Schölkopf et al., 1999): Proposes an algorithm which computes a binary function that is supposed to capture regions in input space where the probability density lives.

- IForest (IF) (Liu et al., 2008): Detects anomalies by explicitly isolating them through recursive partitioning rather than modeling normal behavior.

- HBOS (Goldstein & Dengel, 2012): An unsupervised histogram-based anomaly detection method.

### A.3. Metrics

This subsection presents the evaluation metrics adopted in this work, which can be broadly divided into two categories. The first category consists of label-based metrics, including Affiliated Precision (Aff-P), Affiliated Recall (Aff-R), and Affiliated F1-score (Aff-F) (Huet et al., 2022), as well as Accuracy (Acc), Precision (P), Recall (R), F1-score (F1), Range Precision (R-P), Range Recall (R-R), and Range F1-score (R-F) (Tatbul et al., 2018). The second category comprises score-based metrics, including the Area Under the Precision–Recall Curve (A-P) (Davis & Goadrich, 2006), the Area Under the Receiver Operating Characteristic Curve (A-R) (Fawcett, 2006), the Range Area Under the Precision–Recall Curve (R-A-P), the Range Area Under the Receiver Operating Characteristic Curve (R-A-R), as well as the Volume Under the Precision–Recall Surface (V-PR) and the Volume Under the Receiver Operating Characteristic Surface (V-ROC) (Paparrizos et al., 2022). All these metrics are employed in TeamWork to evaluate the performance of different methods.

### A.4. Implementation Details

We adhere to the evaluation protocol proposed in TFB (Qiu et al., 2024) during testing by disabling the 'drop last' operation, ensuring a fair comparison across all models. We conduct experiments using Pytorch with NVIDIA Tesla-A800-80GB GPUs. We employ the Adam optimizer (Kingma & Ba, 2015) during training. All baseline methods are configured consistently with the optimal settings reported in CATCH (Wu et al., 2025c) and are evaluated using identical hardware. Experiments show that this sample configuration delivers stable and competitive performance, indicating that additional tuning is unnecessary. We have released our model introduction at https://github.com/decisionintelligence/TeamWork.

## B. Proof of theories

### B.1. Mutual Information Equivalence

For the convenience of proof, we let $\Phi = \mathbf{X}_{-x_{i,:}}$, $x_{i,:} = \{\xi_i^1, \xi_i^1, \ldots, \xi_i^T\}$. Based on the definition of entropy and conditional entropy, we can derive another form of entropy reduction index $\Delta_i^{\mathcal{H}}(\mathbf{X}, x_{i,:})$ as follows:

$$
\begin{aligned}
\Delta_i^{\mathcal{H}}(\mathbf{X}, x_{i,:}) &= \mathcal{H}(\Phi) - \mathcal{H}(\Phi|x_{i,:}) \\
&= -\sum_{\phi \in \Phi} p(\phi) \log p(\phi) - \left( -\sum_{\xi_i^t \in x_{i,:}} p(\xi_i^t) \sum_{\phi \in \Phi} p(\phi|\xi_i^t) \log p(\phi|\xi_i^t) \right) \\
&= -\sum_{\phi \in \Phi} p(\phi) \log p(\phi) + \sum_{\xi_i^t \in x_{i,:}} \sum_{\phi \in \Phi} p(\xi_i^t, \phi_i) \log p(\phi|\xi_i^t) \\
&= -\sum_{\xi_i^t \in x_{i,:}} \sum_{\phi \in \Phi} p(\xi_i^t, \phi_i) \log p(\phi) + \sum_{\xi_i^t \in x_{i,:}} \sum_{\phi \in \Phi} p(\xi_i^t, \phi_i) \log p(\phi|\xi_i^t) \\
&= \sum_{\xi_i^t \in x_{i,:}} \sum_{\phi \in \Phi} \left( p(\phi|\xi_i^t) - p(\phi) \right) \\
&= \sum_{\xi_i^t \in x_{i,:}} \sum_{\phi \in \Phi} p(\xi_i^t, \phi_i) \log \frac{p(\phi|\xi_i^t)}{p(\phi)} \\
&= \mathcal{I}(\Phi; x_{i,:})
\end{aligned}
\tag{15}
$$

Therefore, the entropy reduction index is equivalent to mutual information between $\Phi$ and $x_{i,:}$. By substituting $\Phi = \mathbf{X}_{-x_{i,:}}$ back to $\mathcal{I}(\Phi; x_{i,:})$, we can obtain another representation of ERI as

$$
\Delta_i^{\mathcal{H}}(\mathbf{X}, x_{i,:}) = \mathcal{I}(\mathbf{X}_{-x_{i,:}}; x_{i,:}).
\tag{16}
$$

### B.2. Kraskov $k$-NN Estimator for Entropy Reduction

In order to compute the value of mutual information $\mathcal{I}(\mathbf{X}_{-x_{i,:}}; x_{i,:})$ proposed in (16), we should extend the mutual information of double variable to that of multivariate. Firstly, we should introduce the internal correlation of variable set $\mathbf{X}_{-x_{i,:}}$, expressed as $\mathcal{I}(\mathbf{X}_{-x_{i,:}})$, which is expanded as follows:

$$
\begin{aligned}
\mathcal{I}(\mathbf{X}_{-x_{i,:}}) = \sum_{\xi_1^t \in x_{1,:}} \cdots \sum_{\xi_{i-1}^t \in x_{i-1,:}} \sum_{\xi_{i+1}^t \in x_{i+1,:}} \cdots \sum_{\xi_D^t \in x_{D,:}} p(\xi_1^t, \ldots, \xi_{i-1}^t, \xi_{i+1}^t, \ldots, \xi_D^t) \\
\times \log \frac{p(\xi_1^t, \ldots, \xi_{i-1}^t, \xi_{i+1}^t, \ldots, \xi_D^t, \xi_i^t)}{p(\xi_1^t) \cdots, p(\xi_{i-1}^t) p(\xi_{i+1}^t), \cdots, p(\xi_D^t)}.
\end{aligned}
\tag{17}
$$

Then, by considering the dependency between variable $x_{i,:}$ and $\mathbf{X}_{-x_{i,:}}$, the mutual information (17) is re-expressed as

$$
\begin{aligned}
\mathcal{I}(\mathbf{X}_{-x_{i,:}}; x_{i,:}) = \sum_{\xi_1^t \in x_{1,:}} \sum_{\xi_2^t \in x_{2,:}} \cdots \sum_{\xi_D^t \in x_{D,:}} p(\xi_1^t, \xi_2^t, \ldots, \xi_D^t) \\
\times \log \frac{p(\xi_1^t, \xi_2^t, \ldots, \xi_{i-1}^t, \xi_{i+1}^t, \ldots, \xi_D^t, \xi_i^t)}{p(\xi_1^t, \xi_2^t, \ldots, \xi_{i-1}^t, \xi_{i+1}^t, \ldots, \xi_D^t) p(\xi_i^t)}.
\end{aligned}
\tag{18}
$$

where $p(\xi_1^t, \xi_2^t, \ldots, \xi_{i-1}^t, \xi_{i+1}^t, \ldots, \xi_D^t, \xi_i^t)$ denotes the joint probability density function on all variables.

It is obvious that the value of mutual information cannot be computed accurately, thus $k$-nearest neighbor method is proposed to approximate it. From the concept of Kraskov estimator (Kraskov et al., 2004), we find that the high-dimensional variables $\mathbf{X}_{-x_{i,:}} = \{x_{1,:}, \ldots, x_{i-1,:}, x_{i+1,:}, \ldots, x_{D,:}\}$ can be viewed as a whole variable. Hence, the estimation of multivariate mutual information can be solved via the method for mutual information of double variable. Then, define the number of neighbors as $k$ and we should find the distance $\mathcal{V}$ of $k$-th neighbor in the joint space $\mathbf{X}$. Next, the number of points in

subspaces $\mathbf{X}_{-x_{i,:}}$ and $x_{i,:}$ that are less than $\mathcal{V}$ are counted, respectively. Finally, by using these statistics, $\mathcal{I}(\mathbf{X}_{x_{i,:}}; x_{i,:})$ is approximated through the digamma function. Specifically, the value of mutual information is computed as follows:

$$
\begin{aligned}
KE(\Delta_i^{\mathcal{H}}(\mathbf{X}, x_{i,:})) &= \mathcal{I}(\mathbf{X}_{-x_{i,:}}; x_{i,:}) \\
&\approx \psi(k) - \left\langle \psi(n_{\mathbf{X}_{-x_{i,:}}}) + \psi(n_{x_{i,:}}) \right\rangle + \psi(L) \\
&= \psi(k) - \frac{1}{L} \sum_{l=1}^{L} [\psi(n_{\mathbf{X}_{-x_{i,:}}}(l) + 1) + \psi(n_{x_{i,:}}(l) + 1)] + \psi(L).
\end{aligned}
\tag{19}
$$

where $L$ is the number of samples, $\psi(\cdot)$ is the digamma function. $n_{\mathbf{X}_{-x_{i,:}}}$ denotes the number of sample points in the marginal subspace $\mathbf{X}_{-x_{i,:}}$ whose distance from the $l$-th sample is strictly less than $\mathcal{V}$, as well as $n_{x_{i,:}}$.

## C. Broader Connections to Representation Learning

Beyond time-series-specific models, recent progress in signal enhancement, multimodal representation learning, and retrieval-augmented generation has shown that richer representations and cross-view fusion can improve robustness in complex data modeling tasks (Jin et al., 2025; 2026; Fang et al., 2026; 2023; Li et al., 2026b;a). In addition, recent studies on image restoration, manipulation detection, multimodal recommendation, video generation, temporal action localization, vision memory, and diffusion model editing demonstrate the broad effectiveness of deep representation learning and alignment mechanisms (Zhang et al., 2024; 2025b; Chen et al., 2025; Qi et al., 2025; Ma et al., 2024; 2025; Zhang et al., 2025a;c; Yu et al., 2025c; Lu et al., 2024; 2023). Although these works target different data modalities, they further motivate the general principle of learning informative, role-specific, and complementary representations from heterogeneous signals.

## D. Additional Experimental Results

In this section, we present the performance of TeamWork and 16 baseline methods on additional evaluation metrics. Specifically, Tables 6, 5, 7, 8, 9, and 10 present the comparative evaluation results across the following metrics: (AUC-ROC, R-AUC-ROC, VUS-ROC), (Accuracy), (AUC-PR, R-AUC-PR, VUS-PR), (Precision, Recall, F1-score), (Range-Recall, Range-Precision, Range-F1-score), and (Affiliated-Precision, Affiliated-Recall, Affiliated-F1-score), respectively.

*Table 5.* Performance comparison of various methods on three metrics: A-R (AUC-ROC), R-A-R (R-AUC-ROC), and V-ROC (VUS-ROC). Best results are in **bold**, second-best are underlined.

| Dataset | Metric | TeamWork | CrossAD | CATCH | Modern | iTrans | DualTF | ATrans | DC | TsNet | Patch | DLin | NLin | AE | Ocsvm | IF | PCA | HBOS |
|---|---|---|---|---|---|---|---|---|---|---|---|---|---|---|---|---|---|---|
| GECCO | A-R | **0.992** | 0.985 | 0.970 | 0.952 | 0.795 | 0.714 | 0.516 | 0.555 | 0.954 | 0.949 | 0.947 | 0.936 | 0.769 | 0.804 | 0.619 | 0.711 | 0.557 |
| | R-A-R | **0.993** | 0.990 | 0.990 | 0.978 | 0.884 | 0.725 | 0.501 | 0.609 | 0.977 | 0.984 | 0.987 | 0.977 | 0.634 | 0.757 | 0.680 | 0.598 | 0.510 |
| | V-ROC | **0.993** | 0.989 | 0.987 | 0.975 | 0.871 | 0.717 | 0.503 | 0.593 | 0.974 | 0.979 | 0.982 | 0.971 | 0.637 | 0.756 | 0.677 | 0.595 | 0.503 |
| Genesis | A-R | **0.994** | 0.812 | 0.974 | 0.676 | 0.690 | 0.937 | 0.947 | 0.659 | 0.913 | 0.685 | 0.696 | 0.755 | 0.931 | 0.733 | 0.549 | 0.815 | 0.897 |
| | R-A-R | **0.995** | 0.867 | 0.981 | 0.727 | 0.797 | 0.975 | 0.976 | 0.744 | 0.919 | 0.737 | 0.741 | 0.791 | 0.917 | 0.734 | 0.693 | 0.822 | 0.819 |
| | V-ROC | **0.995** | 0.859 | 0.978 | 0.729 | 0.773 | 0.971 | 0.970 | 0.730 | 0.913 | 0.728 | 0.735 | 0.787 | 0.916 | 0.733 | 0.660 | 0.816 | 0.827 |
| CalIt2 | A-R | **0.850** | 0.786 | 0.838 | 0.676 | 0.791 | 0.574 | 0.533 | 0.527 | 0.771 | 0.808 | 0.752 | 0.695 | 0.767 | 0.804 | 0.775 | 0.790 | 0.798 |
| | R-A-R | **0.866** | 0.815 | 0.854 | 0.726 | 0.817 | 0.666 | 0.523 | 0.505 | 0.807 | 0.830 | 0.794 | 0.743 | 0.798 | 0.840 | 0.818 | 0.796 | 0.847 |
| | V-ROC | **0.859** | 0.805 | 0.848 | 0.716 | 0.809 | 0.630 | 0.517 | 0.503 | 0.796 | 0.824 | 0.781 | 0.728 | 0.789 | 0.828 | 0.802 | 0.786 | 0.831 |
| Credit | A-R | **0.963** | 0.948 | 0.958 | 0.957 | 0.934 | 0.703 | 0.552 | 0.504 | 0.957 | 0.957 | 0.954 | 0.948 | 0.909 | 0.953 | 0.860 | 0.871 | 0.951 |
| | R-A-R | **0.935** | 0.671 | 0.921 | 0.919 | 0.882 | 0.535 | 0.379 | 0.483 | 0.919 | 0.918 | 0.914 | 0.902 | 0.843 | 0.914 | 0.809 | 0.784 | 0.904 |
| | V-ROC | **0.922** | 0.913 | 0.917 | 0.908 | 0.881 | 0.575 | 0.398 | 0.483 | 0.914 | 0.911 | 0.908 | 0.897 | 0.840 | 0.905 | 0.818 | 0.805 | 0.896 |
| MSL | A-R | **0.681** | 0.594 | 0.664 | 0.633 | 0.611 | 0.576 | 0.508 | 0.507 | 0.613 | 0.637 | 0.624 | 0.592 | 0.562 | 0.524 | 0.524 | 0.552 | 0.574 |
| | R-A-R | 0.735 | 0.465 | **0.747** | 0.708 | 0.686 | 0.662 | 0.529 | 0.596 | 0.701 | 0.720 | 0.703 | 0.681 | 0.635 | 0.594 | 0.575 | 0.631 | 0.643 |
| | V-ROC | 0.725 | 0.664 | **0.735** | 0.701 | 0.678 | 0.652 | 0.527 | 0.587 | 0.692 | 0.712 | 0.695 | 0.672 | 0.628 | 0.590 | 0.571 | 0.622 | 0.635 |
| SMAP | A-R | **0.592** | 0.440 | 0.431 | 0.434 | 0.409 | 0.465 | 0.501 | 0.500 | 0.442 | 0.441 | 0.391 | 0.437 | 0.395 | 0.393 | 0.487 | 0.396 | 0.585 |
| | R-A-R | 0.593 | **0.603** | 0.459 | 0.470 | 0.441 | 0.504 | 0.507 | 0.507 | 0.482 | 0.479 | 0.427 | 0.472 | 0.430 | 0.430 | 0.499 | 0.420 | 0.565 |
| | V-ROC | **0.591** | 0.463 | 0.457 | 0.468 | 0.439 | 0.502 | 0.507 | 0.506 | 0.480 | 0.478 | 0.427 | 0.471 | 0.429 | 0.428 | 0.499 | 0.419 | 0.566 |
| PSM | A-R | **0.720** | 0.615 | 0.652 | 0.593 | 0.592 | 0.600 | 0.514 | 0.501 | 0.592 | 0.586 | 0.580 | 0.585 | 0.650 | 0.619 | 0.542 | 0.648 | 0.620 |
| | R-A-R | 0.679 | **0.768** | 0.640 | 0.588 | 0.589 | 0.507 | 0.453 | 0.489 | 0.593 | 0.586 | 0.579 | 0.585 | 0.587 | 0.530 | 0.543 | 0.584 | 0.572 |
| | V-ROC | **0.679** | 0.606 | 0.639 | 0.589 | 0.588 | 0.507 | 0.451 | 0.479 | 0.593 | 0.585 | 0.579 | 0.585 | 0.589 | 0.532 | 0.542 | 0.585 | 0.575 |
| SMD | A-R | **0.825** | 0.748 | 0.811 | 0.722 | 0.745 | 0.631 | 0.508 | 0.502 | 0.727 | 0.736 | 0.728 | 0.738 | 0.774 | 0.602 | 0.664 | 0.679 | 0.626 |
| | R-A-R | **0.831** | 0.768 | 0.800 | 0.743 | 0.762 | 0.594 | 0.500 | 0.505 | 0.747 | 0.760 | 0.754 | 0.762 | 0.783 | 0.579 | 0.679 | 0.656 | 0.597 |
| | V-ROC | **0.830** | 0.765 | 0.797 | 0.742 | 0.761 | 0.592 | 0.499 | 0.505 | 0.746 | 0.758 | 0.751 | 0.760 | 0.782 | 0.578 | 0.678 | 0.655 | 0.597 |

*Table 6.* Accuracy (Acc) comparison across baselines and datasets. Best results in **bold**, second-best underlined.

| Method | GECCO | Genesis | CalIt2 | Credit | MSL | SMAP | PSM | SMD |
|---|---|---|---|---|---|---|---|---|
| Metric | Acc | Acc | Acc | Acc | Acc | Acc | Acc | Acc |
| HBOS | 0.590 | 0.771 | 0.886 | 0.847 | 0.816 | 0.821 | 0.736 | 0.841 |
| PCA | 0.240 | 0.888 | 0.884 | 0.868 | 0.812 | 0.815 | 0.682 | 0.860 |
| IF | **0.989** | 0.983 | **0.968** | **0.996** | 0.890 | 0.870 | 0.723 | 0.958 |
| Ocsvm | 0.055 | 0.424 | 0.867 | 0.893 | 0.818 | 0.813 | **0.737** | 0.865 |
| AE | 0.888 | 0.987 | 0.960 | 0.994 | 0.891 | 0.872 | 0.715 | **0.959** |
| NLin | 0.987 | 0.972 | 0.844 | 0.980 | 0.872 | 0.685 | 0.728 | 0.943 |
| DLin | 0.981 | 0.987 | 0.887 | 0.981 | 0.858 | 0.684 | 0.726 | 0.943 |
| Patch | **0.989** | 0.987 | 0.889 | 0.980 | 0.873 | 0.725 | 0.729 | 0.934 |
| TsNet | 0.984 | 0.991 | 0.946 | 0.981 | 0.855 | 0.680 | 0.726 | 0.931 |
| DC | 0.979 | 0.991 | 0.841 | 0.716 | **0.892** | 0.833 | 0.615 | 0.774 |
| ATrans | 0.983 | 0.931 | 0.944 | 0.847 | 0.891 | 0.855 | 0.719 | 0.952 |
| DualTF | 0.612 | 0.970 | 0.910 | 0.688 | 0.891 | 0.663 | 0.697 | 0.863 |
| iTrans | 0.981 | 0.986 | 0.930 | 0.951 | 0.854 | 0.684 | 0.728 | 0.914 |
| Modern | 0.984 | 0.965 | 0.889 | 0.981 | 0.857 | 0.686 | 0.729 | 0.931 |
| CATCH | 0.984 | 0.992 | 0.946 | 0.981 | 0.853 | 0.724 | 0.730 | 0.918 |
| CrossAD | 0.983 | 0.976 | 0.753 | 0.951 | 0.816 | 0.872 | **0.740** | 0.865 |
| **TeamWork** | **0.989** | **0.996** | 0.799 | 0.980 | 0.844 | **0.999** | 0.721 | 0.919 |

*Table 7.* A-P (AUC-PR), R-A-P (R-AUC-PR), and V-PR (VUS-PR) performance comparison on all datasets. Best results are in **bold**, second-best are underlined.

| Dataset | Metric | TeamWork | CrossAD | CATCH | Modern | iTrans | DualTF | ATrans | DC | TsNet | Patch | DLin | NLin | AE | Ocsvm | IF | PCA | HBOS |
|---|---|---|---|---|---|---|---|---|---|---|---|---|---|---|---|---|---|---|
| **GECCO** | A-P | 0.440 | **0.494** | 0.418 | 0.447 | 0.096 | 0.130 | 0.013 | 0.012 | 0.410 | 0.400 | 0.349 | 0.304 | 0.206 | 0.039 | 0.052 | 0.234 | 0.199 |
| | R-A-P | **0.537** | 0.497 | 0.473 | 0.459 | 0.134 | 0.050 | 0.022 | 0.020 | 0.428 | 0.444 | 0.416 | 0.372 | 0.033 | 0.098 | 0.085 | 0.047 | 0.032 |
| | V-PR | **0.531** | 0.491 | 0.465 | 0.461 | 0.128 | 0.049 | 0.022 | 0.019 | 0.429 | 0.439 | 0.406 | 0.363 | 0.033 | 0.101 | 0.083 | 0.046 | 0.033 |
| **Genesis** | A-P | **0.420** | 0.020 | 0.249 | 0.015 | 0.019 | 0.051 | 0.058 | 0.010 | 0.036 | 0.013 | 0.017 | 0.011 | 0.055 | 0.007 | 0.005 | 0.011 | 0.059 |
| | R-A-P | **0.596** | 0.025 | 0.384 | 0.015 | 0.021 | 0.101 | 0.101 | 0.012 | 0.038 | 0.013 | 0.016 | 0.013 | 0.047 | 0.506 | 0.011 | 0.015 | 0.090 |
| | V-PR | **0.581** | 0.024 | 0.371 | 0.015 | 0.020 | 0.097 | 0.095 | 0.012 | 0.037 | 0.013 | 0.016 | 0.014 | 0.047 | 0.506 | 0.010 | 0.015 | 0.087 |
| **CalIt2** | A-P | **0.148** | 0.093 | 0.114 | 0.054 | 0.106 | 0.057 | 0.045 | 0.035 | 0.078 | 0.116 | 0.097 | 0.054 | 0.084 | 0.095 | 0.080 | 0.073 | 0.080 |
| | R-A-P | **0.160** | 0.105 | 0.124 | 0.070 | 0.111 | 0.088 | 0.089 | 0.087 | 0.092 | 0.115 | 0.097 | 0.073 | 0.096 | 0.109 | 0.095 | 0.106 | 0.113 |
| | V-PR | **0.156** | 0.103 | 0.121 | 0.070 | 0.110 | 0.082 | 0.083 | 0.083 | 0.090 | 0.115 | 0.095 | 0.072 | 0.097 | 0.109 | 0.091 | 0.103 | 0.108 |
| **Credit** | A-P | 0.050 | 0.077 | 0.101 | 0.088 | 0.042 | 0.023 | 0.007 | 0.002 | 0.091 | 0.089 | 0.081 | 0.087 | 0.040 | 0.053 | 0.074 | 0.029 | **0.173** |
| | R-A-P | 0.038 | 0.039 | 0.053 | 0.054 | 0.024 | 0.009 | 0.002 | 0.002 | 0.056 | 0.054 | 0.053 | 0.053 | 0.020 | 0.038 | 0.040 | 0.009 | **0.081** |
| | V-PR | 0.036 | 0.041 | 0.051 | 0.051 | 0.024 | 0.011 | 0.002 | 0.002 | 0.053 | 0.051 | 0.050 | 0.051 | 0.020 | 0.037 | 0.039 | 0.017 | **0.076** |
| **MSL** | A-P | **0.180** | 0.160 | 0.167 | 0.146 | 0.151 | 0.156 | 0.107 | 0.107 | 0.146 | 0.157 | 0.147 | 0.140 | 0.148 | 0.153 | 0.114 | 0.157 | 0.132 |
| | R-A-P | **0.282** | 0.221 | 0.260 | 0.224 | 0.227 | 0.220 | 0.165 | 0.157 | 0.231 | 0.242 | 0.225 | 0.222 | 0.200 | 0.185 | 0.174 | 0.203 | 0.190 |
| | V-PR | **0.277** | 0.218 | 0.256 | 0.220 | 0.224 | 0.218 | 0.162 | 0.156 | 0.227 | 0.237 | 0.221 | 0.218 | 0.199 | 0.185 | 0.173 | 0.200 | 0.189 |
| **SMAP** | A-P | **0.392** | 0.111 | 0.115 | 0.110 | 0.114 | 0.122 | 0.128 | 0.128 | 0.119 | 0.111 | 0.104 | 0.110 | 0.101 | 0.102 | 0.122 | 0.104 | 0.148 |
| | R-A-P | **0.398** | 0.125 | 0.129 | 0.124 | 0.126 | 0.140 | 0.148 | 0.148 | 0.136 | 0.129 | 0.118 | 0.126 | 0.116 | 0.117 | 0.135 | 0.118 | 0.165 |
| | V-PR | **0.399** | 0.125 | 0.129 | 0.125 | 0.126 | 0.139 | 0.149 | 0.147 | 0.136 | 0.129 | 0.119 | 0.126 | 0.116 | 0.117 | 0.136 | 0.118 | 0.166 |
| **PSM** | A-P | **0.536** | 0.422 | 0.434 | 0.385 | 0.383 | 0.411 | 0.298 | 0.281 | 0.391 | 0.378 | 0.371 | 0.376 | 0.465 | 0.418 | 0.334 | 0.468 | 0.394 |
| | R-A-P | **0.489** | 0.419 | 0.435 | 0.383 | 0.386 | 0.353 | 0.293 | 0.283 | 0.395 | 0.379 | 0.372 | 0.378 | 0.420 | 0.369 | 0.334 | 0.423 | 0.369 |
| | V-PR | **0.489** | 0.421 | 0.436 | 0.384 | 0.387 | 0.354 | 0.293 | 0.283 | 0.395 | 0.380 | 0.373 | 0.379 | 0.420 | 0.370 | 0.334 | 0.423 | 0.370 |
| **SMD** | A-P | **0.215** | 0.149 | 0.172 | 0.130 | 0.146 | 0.069 | 0.046 | 0.043 | 0.141 | 0.147 | 0.139 | 0.141 | 0.188 | 0.104 | 0.122 | 0.128 | 0.145 |
| | R-A-P | **0.219** | 0.145 | 0.159 | 0.130 | 0.145 | 0.070 | 0.055 | 0.046 | 0.140 | 0.152 | 0.145 | 0.145 | 0.182 | 0.080 | 0.099 | 0.109 | 0.088 |
| | V-PR | **0.219** | 0.144 | 0.159 | 0.130 | 0.145 | 0.070 | 0.054 | 0.046 | 0.140 | 0.152 | 0.144 | 0.145 | 0.181 | 0.081 | 0.099 | 0.109 | 0.088 |

*Table 8.* Precision (P), Recall (R), and F1-score performance comparison of various methods on all datasets. Best results are in **bold**, second-best are underlined.

| Dataset | Metric | TeamWork | CrossAD | CATCH | Modern | iTrans | DualTF | ATrans | DC | TsNet | Patch | DLin | NLin | AE | Ocsvm | IF | PCA | HBOS |
|---|---|---|---|---|---|---|---|---|---|---|---|---|---|---|---|---|---|---|
| **GECCO** | P | **0.495** | 0.362 | 0.380 | 0.373 | 0.173 | 0.009 | 0.012 | 0.008 | 0.379 | 0.475 | 0.332 | 0.384 | 0.021 | 0.001 | 0.214 | 0.014 | 0.014 |
| | R | 0.626 | 0.785 | 0.818 | 0.779 | 0.207 | 0.340 | 0.008 | 0.008 | 0.804 | 0.585 | 0.781 | 0.460 | 0.215 | **0.993** | 0.012 | 0.988 | 0.542 |
| | F1 | **0.553** | 0.496 | 0.518 | 0.504 | 0.189 | 0.018 | 0.010 | 0.008 | 0.516 | 0.524 | 0.466 | 0.418 | 0.039 | 0.022 | 0.023 | 0.027 | 0.027 |
| **Genesis** | P | **0.429** | 0.030 | 0.116 | 0.015 | 0.065 | 0.066 | 0.053 | 0.016 | 0.119 | 0.075 | 0.055 | 0.013 | 0.047 | 0.007 | 0.006 | 0.017 | 0.017 |
| | R | 0.180 | 0.160 | 0.160 | 0.120 | 0.180 | 0.500 | 0.960 | 0.020 | 0.200 | 0.200 | 0.140 | 0.080 | 0.120 | **1.000** | 0.020 | 0.480 | 0.980 |
| | F1 | **0.254** | 0.051 | 0.134 | 0.027 | 0.095 | 0.116 | 0.100 | 0.018 | 0.149 | 0.109 | 0.079 | 0.022 | 0.068 | 0.014 | 0.009 | 0.033 | 0.033 |
| **CallIt2** | P | 0.101 | 0.080 | **0.138** | 0.074 | 0.124 | 0.073 | 0.085 | 0.034 | 0.104 | 0.091 | 0.083 | 0.066 | 0.067 | 0.089 | 0.125 | 0.084 | 0.075 |
| | R | **0.743** | 0.703 | 0.162 | 0.243 | 0.230 | 0.176 | 0.095 | 0.162 | 0.108 | 0.311 | 0.284 | 0.324 | 0.027 | 0.378 | 0.014 | 0.297 | 0.257 |
| | F1 | **0.179** | 0.143 | 0.149 | 0.114 | 0.161 | 0.103 | 0.090 | 0.056 | 0.106 | 0.141 | 0.128 | 0.109 | 0.038 | 0.144 | 0.024 | 0.131 | 0.117 |
| **Credit** | P | 0.060 | 0.026 | 0.059 | 0.058 | 0.026 | 0.003 | 0.001 | 0.001 | 0.058 | 0.058 | 0.057 | 0.057 | 0.047 | 0.013 | **0.156** | 0.009 | 0.009 |
| | R | 0.807 | 0.834 | 0.758 | 0.740 | 0.843 | 0.596 | 0.139 | 0.238 | 0.740 | 0.749 | 0.726 | 0.744 | 0.135 | 0.901 | 0.283 | 0.731 | **0.915** |
| | F1 | 0.112 | 0.051 | 0.110 | 0.107 | 0.051 | 0.006 | 0.003 | 0.003 | 0.107 | 0.105 | 0.105 | 0.107 | 0.069 | 0.026 | **0.201** | 0.017 | 0.018 |
| **MSL** | P | 0.157 | 0.147 | 0.185 | 0.166 | 0.158 | **0.248** | 0.143 | 0.181 | 0.166 | 0.194 | 0.173 | 0.193 | 0.219 | 0.128 | 0.180 | 0.130 | 0.127 |
| | R | 0.110 | **0.155** | 0.117 | 0.090 | 0.089 | 0.019 | 0.008 | 0.008 | 0.093 | 0.066 | 0.093 | 0.067 | 0.014 | 0.125 | 0.014 | 0.137 | 0.128 |
| | F1 | 0.129 | **0.151** | 0.143 | 0.117 | 0.114 | 0.035 | 0.015 | 0.016 | 0.119 | 0.098 | 0.121 | 0.100 | 0.026 | 0.126 | 0.025 | 0.133 | 0.127 |
| **SMAP** | P | 0.057 | **0.465** | 0.087 | 0.084 | 0.083 | 0.119 | 0.142 | 0.135 | 0.089 | 0.093 | 0.086 | 0.086 | 0.148 | 0.078 | 0.092 | 0.084 | 0.092 |
| | R | 0.186 | 0.004 | 0.122 | 0.147 | 0.146 | **0.256** | 0.027 | 0.057 | 0.163 | 0.132 | 0.153 | 0.152 | 0.000 | 0.042 | 0.002 | 0.045 | 0.045 |
| | F1 | 0.108 | 0.007 | 0.102 | 0.107 | 0.105 | **0.163** | 0.046 | 0.080 | 0.115 | 0.110 | 0.110 | 0.110 | 0.001 | 0.055 | 0.004 | 0.059 | 0.060 |
| **PSM** | P | 0.494 | 0.684 | 0.624 | 0.653 | 0.600 | 0.461 | 0.396 | 0.278 | 0.591 | 0.653 | 0.658 | 0.644 | 0.444 | 0.548 | **0.802** | 0.427 | 0.544 |
| | R | 0.169 | 0.119 | 0.064 | 0.047 | 0.058 | **0.559** | 0.025 | 0.243 | 0.048 | 0.047 | 0.025 | 0.047 | 0.110 | 0.299 | 0.004 | 0.421 | 0.302 |
| | F1 | 0.251 | 0.203 | 0.116 | 0.089 | 0.105 | **0.506** | 0.046 | 0.259 | 0.088 | 0.088 | 0.047 | 0.087 | 0.176 | 0.387 | 0.007 | 0.424 | 0.389 |
| **SMD** | P | 0.196 | 0.134 | 0.194 | 0.151 | 0.162 | 0.091 | 0.112 | 0.042 | 0.176 | 0.185 | 0.197 | 0.201 | **0.706** | 0.107 | 0.454 | 0.117 | 0.095 |
| | R | 0.307 | **0.410** | 0.305 | 0.145 | 0.255 | 0.257 | 0.022 | 0.201 | 0.181 | 0.173 | 0.124 | 0.126 | 0.007 | 0.304 | 0.020 | 0.359 | 0.332 |
| | F1 | **0.239** | 0.202 | 0.237 | 0.148 | 0.198 | 0.135 | 0.037 | 0.069 | 0.178 | 0.179 | 0.152 | 0.155 | 0.014 | 0.158 | 0.039 | 0.176 | 0.148 |

*Table 9.* R-R (Recall-Recall), R-P (Recall-Precision), and R-F (Recall-F1) performance comparison on all datasets. Best results are in **bold**, second-best are underlined.

| Dataset | Metric | TeamWork | CrossAD | CATCH | Modern | iTrans | DualTF | ATrans | DC | TsNet | Patch | DLin | NLin | AE | Ocsvm | IF | PCA | HBOS |
|---|---|---|---|---|---|---|---|---|---|---|---|---|---|---|---|---|---|---|
| **GECCO** | R-R | 0.635 | 0.804 | 0.795 | 0.644 | 0.266 | 0.146 | 0.063 | 0.040 | 0.782 | 0.366 | 0.790 | 0.274 | 0.188 | 0.953 | 0.030 | **0.986** | 0.361 |
| | R-P | **0.312** | 0.074 | 0.065 | 0.086 | 0.111 | 0.039 | 0.013 | 0.008 | 0.053 | 0.289 | 0.042 | 0.296 | 0.021 | 0.006 | 0.214 | 0.041 | 0.016 |
| | R-F | **0.419** | 0.136 | 0.119 | 0.152 | 0.156 | 0.062 | 0.022 | 0.013 | 0.099 | 0.323 | 0.080 | 0.285 | 0.038 | 0.013 | 0.052 | 0.079 | 0.031 |
| **Genesis** | R-R | 0.233 | 0.223 | 0.497 | 0.174 | 0.507 | 0.550 | 0.855 | 0.079 | 0.211 | 0.385 | 0.180 | 0.162 | 0.356 | **1.000** | 0.077 | 0.325 | 0.861 |
| | R-P | **0.333** | 0.042 | 0.119 | 0.014 | 0.057 | 0.030 | 0.005 | 0.016 | 0.086 | 0.067 | 0.059 | 0.013 | 0.047 | 0.003 | 0.006 | 0.035 | 0.002 |
| | R-F | **0.274** | 0.070 | 0.192 | 0.026 | 0.102 | 0.058 | 0.010 | 0.027 | 0.122 | 0.114 | 0.089 | 0.024 | 0.083 | 0.005 | 0.011 | 0.063 | 0.005 |
| **CallIt2** | R-R | **0.774** | 0.714 | 0.278 | 0.318 | 0.344 | 0.209 | 0.161 | 0.205 | 0.161 | 0.424 | 0.315 | 0.325 | 0.055 | 0.359 | 0.031 | 0.280 | 0.309 |
| | R-P | 0.074 | 0.079 | **0.139** | 0.069 | 0.113 | 0.056 | 0.105 | 0.031 | 0.095 | 0.077 | 0.091 | 0.064 | 0.067 | 0.096 | 0.125 | 0.099 | 0.086 |
| | R-F | 0.135 | 0.143 | **0.185** | 0.114 | 0.170 | 0.088 | 0.127 | 0.053 | 0.120 | 0.131 | 0.142 | 0.107 | 0.060 | 0.152 | 0.049 | 0.147 | 0.134 |
| **Credit** | R-R | 0.789 | 0.819 | 0.735 | 0.716 | 0.828 | 0.564 | 0.150 | 0.234 | 0.716 | 0.725 | 0.701 | 0.721 | 0.135 | 0.892 | 0.292 | 0.709 | **0.907** |
| | R-P | 0.057 | 0.025 | 0.054 | 0.052 | 0.024 | 0.002 | 0.003 | 0.002 | 0.051 | 0.052 | 0.049 | 0.052 | 0.047 | 0.012 | **0.156** | 0.008 | 0.009 |
| | R-F | 0.106 | 0.049 | 0.101 | 0.097 | 0.047 | 0.004 | 0.007 | 0.003 | 0.096 | 0.096 | 0.092 | 0.097 | 0.070 | 0.023 | **0.204** | 0.015 | 0.017 |
| **MSL** | R-R | 0.233 | **0.261** | 0.241 | 0.194 | 0.182 | 0.049 | 0.150 | 0.126 | 0.224 | 0.176 | 0.202 | 0.162 | 0.110 | 0.202 | 0.112 | 0.237 | 0.199 |
| | R-P | 0.149 | 0.120 | 0.150 | 0.129 | 0.125 | **0.267** | 0.143 | 0.179 | 0.130 | 0.136 | 0.130 | 0.137 | 0.219 | 0.098 | 0.180 | 0.113 | 0.119 |
| | R-F | 0.182 | 0.165 | **0.185** | 0.155 | 0.148 | 0.084 | 0.146 | 0.148 | 0.164 | 0.153 | 0.158 | 0.149 | 0.146 | 0.132 | 0.138 | 0.153 | 0.149 |
| **SMAP** | R-R | 0.215 | 0.064 | 0.210 | **0.264** | 0.202 | 0.226 | 0.192 | 0.191 | 0.320 | 0.250 | 0.238 | 0.254 | 0.051 | 0.134 | 0.092 | 0.137 | 0.128 |
| | R-P | 0.020 | **0.162** | 0.078 | 0.083 | 0.079 | 0.110 | 0.123 | 0.136 | 0.093 | 0.081 | 0.089 | 0.082 | 0.148 | 0.079 | 0.092 | 0.086 | 0.075 |
| | R-F | 0.039 | 0.092 | 0.113 | 0.126 | 0.113 | 0.148 | 0.150 | **0.159** | 0.144 | 0.122 | 0.130 | 0.124 | 0.076 | 0.100 | 0.092 | 0.105 | 0.094 |
| **PSM** | R-R | **0.677** | 0.489 | 0.450 | 0.376 | 0.470 | 0.410 | 0.133 | 0.211 | 0.455 | 0.370 | 0.399 | 0.359 | 0.230 | 0.133 | 0.139 | 0.505 | 0.265 |
| | R-P | 0.423 | 0.553 | 0.557 | 0.553 | 0.533 | 0.456 | 0.374 | 0.274 | 0.537 | 0.557 | 0.584 | 0.587 | 0.444 | 0.385 | **0.802** | 0.459 | 0.467 |
| | R-F | **0.521** | 0.519 | 0.498 | 0.448 | 0.499 | 0.432 | 0.197 | 0.238 | 0.493 | 0.444 | 0.474 | 0.446 | 0.303 | 0.198 | 0.237 | 0.481 | 0.338 |
| **SMD** | R-R | 0.514 | **0.625** | 0.478 | 0.378 | 0.360 | 0.187 | 0.126 | 0.276 | 0.385 | 0.426 | 0.400 | 0.384 | 0.131 | 0.323 | 0.122 | 0.404 | 0.297 |
| | R-P | 0.109 | 0.059 | 0.095 | 0.092 | 0.115 | 0.055 | 0.101 | 0.042 | 0.110 | 0.121 | 0.124 | 0.131 | **0.706** | 0.067 | 0.454 | 0.123 | 0.062 |
| | R-F | 0.180 | 0.107 | 0.158 | 0.148 | 0.175 | 0.085 | 0.112 | 0.073 | 0.171 | 0.189 | 0.189 | **0.196** | 0.143 | 0.111 | 0.192 | 0.188 | 0.103 |

*Table 10.* Average Aff-P (Affiliated-Precision), Aff-R (Affiliated-Recall), and Aff-F (Affiliated-F1score) accuracy measures for all datasets. Best results are in **bold**, second-best are underlined.

| Dataset | Metric | TeamWork | CrossAD | CATCH | Modern | iTrans | DualTF | ATrans | DC | TsNet | Patch | DLin | NLin | AE | Ocsvm | IF | PCA | HBOS |
|---|---|---|---|---|---|---|---|---|---|---|---|---|---|---|---|---|---|---|
| **GECCO** | Aff-P | **0.883** | 0.810 | 0.832 | 0.808 | 0.735 | 0.633 | 0.690 | 0.567 | 0.810 | 0.831 | 0.808 | 0.793 | 0.836 | 0.499 | 0.647 | 0.646 | 0.620 |
| | Aff-R | 0.992 | 0.997 | 0.998 | 0.998 | 0.979 | 0.786 | 0.903 | 0.872 | 0.997 | 0.995 | 0.997 | 0.992 | 0.810 | **1.000** | 0.315 | **1.000** | 0.827 |
| | Aff-F | **0.934** | 0.894 | 0.908 | 0.893 | 0.839 | 0.701 | 0.782 | 0.687 | 0.894 | 0.906 | 0.893 | 0.882 | 0.823 | 0.666 | 0.424 | 0.785 | 0.708 |
| **Genesis** | Aff-P | **0.896** | 0.762 | 0.835 | 0.728 | 0.822 | 0.683 | 0.749 | 0.659 | 0.780 | 0.763 | 0.772 | 0.724 | 0.759 | 0.512 | 0.673 | 0.691 | 0.564 |
| | Aff-R | 0.962 | 0.966 | 0.966 | 0.974 | 0.972 | 0.996 | **1.000** | 0.943 | 0.968 | 0.974 | 0.959 | 0.971 | 0.976 | **1.000** | 0.951 | 0.991 | **1.000** |
| | Aff-F | **0.928** | 0.852 | 0.896 | 0.833 | 0.891 | 0.810 | 0.856 | 0.776 | 0.864 | 0.856 | 0.856 | 0.829 | 0.854 | 0.677 | 0.788 | 0.814 | 0.721 |
| **CalIt2** | Aff-P | **0.745** | 0.649 | 0.742 | 0.650 | 0.703 | 0.617 | 0.645 | 0.571 | 0.691 | 0.667 | 0.668 | 0.616 | 0.560 | 0.652 | 0.539 | 0.688 | 0.620 |
| | Aff-R | 0.993 | **0.996** | 0.955 | 0.975 | 0.963 | 0.959 | 0.838 | 0.894 | 0.932 | 0.976 | 0.976 | 0.984 | 0.617 | 0.982 | 0.321 | 0.869 | 0.969 |
| | Aff-F | **0.852** | 0.786 | 0.835 | 0.780 | 0.812 | 0.751 | 0.729 | 0.697 | 0.794 | 0.793 | 0.793 | 0.757 | 0.587 | 0.783 | 0.402 | 0.768 | 0.756 |
| **Credit** | Aff-P | 0.622 | 0.582 | 0.618 | 0.611 | 0.559 | 0.513 | 0.521 | 0.488 | 0.610 | 0.612 | 0.605 | 0.608 | 0.560 | 0.556 | **0.658** | 0.564 | 0.533 |
| | Aff-R | 0.964 | 0.984 | 0.956 | 0.952 | 0.984 | 0.935 | 0.865 | 0.898 | 0.953 | 0.955 | 0.948 | 0.952 | 0.562 | 0.997 | 0.612 | 0.959 | **0.999** |
| | Aff-F | **0.756** | 0.731 | 0.750 | 0.744 | 0.713 | 0.663 | 0.650 | 0.632 | 0.744 | 0.746 | 0.738 | 0.742 | 0.561 | 0.714 | 0.634 | 0.710 | 0.695 |
| **MSL** | Aff-P | **0.607** | 0.550 | 0.599 | 0.578 | 0.566 | 0.562 | 0.549 | 0.576 | 0.589 | 0.584 | 0.577 | 0.584 | 0.521 | 0.497 | 0.502 | 0.538 | 0.520 |
| | Aff-R | 0.962 | 0.978 | 0.966 | 0.975 | 0.951 | 0.618 | 0.933 | 0.874 | 0.973 | 0.952 | 0.975 | 0.948 | 0.781 | 0.902 | 0.697 | 0.914 | **0.982** |
| | Aff-F | **0.744** | 0.704 | 0.740 | 0.726 | 0.710 | 0.588 | 0.692 | 0.694 | 0.734 | 0.724 | 0.725 | 0.723 | 0.625 | 0.641 | 0.584 | 0.678 | 0.680 |
| **SMAP** | Aff-P | **0.761** | 0.621 | 0.440 | 0.457 | 0.438 | 0.509 | 0.532 | 0.519 | 0.473 | 0.466 | 0.447 | 0.451 | 0.466 | 0.392 | 0.425 | 0.399 | 0.399 |
| | Aff-R | **1.000** | 0.547 | 0.841 | 0.947 | 0.848 | 0.997 | 0.988 | 0.990 | 0.975 | 0.969 | 0.945 | 0.942 | 0.399 | 0.703 | 0.645 | 0.687 | 0.703 |
| | Aff-F | **0.864** | 0.582 | 0.578 | 0.616 | 0.577 | 0.674 | 0.692 | 0.681 | 0.637 | 0.629 | 0.607 | 0.610 | 0.430 | 0.503 | 0.512 | 0.505 | 0.509 |
| **PSM** | Aff-P | 0.771 | 0.779 | 0.808 | 0.734 | 0.765 | 0.622 | 0.600 | 0.538 | 0.762 | 0.739 | 0.777 | 0.762 | 0.776 | 0.652 | **0.904** | 0.712 | 0.621 |
| | Aff-R | 0.965 | 0.920 | 0.918 | 0.941 | **0.966** | 0.868 | 0.871 | 0.932 | 0.939 | 0.948 | 0.993 | 0.942 | 0.649 | 0.447 | 0.472 | 0.692 | 0.700 |
| | Aff-F | 0.857 | 0.843 | **0.859** | 0.825 | 0.854 | 0.725 | 0.710 | 0.682 | 0.842 | 0.831 | 0.831 | 0.843 | 0.707 | 0.531 | 0.620 | 0.702 | 0.658 |
| **SMD** | Aff-P | 0.773 | 0.754 | 0.773 | 0.755 | 0.736 | 0.527 | 0.607 | 0.510 | 0.745 | 0.748 | 0.761 | 0.762 | **0.889** | 0.649 | 0.801 | 0.680 | 0.557 |
| | Aff-R | 0.952 | 0.979 | 0.938 | 0.948 | 0.943 | 0.956 | 0.895 | **0.998** | 0.938 | 0.970 | 0.940 | 0.946 | 0.291 | 0.866 | 0.513 | 0.807 | 0.722 |
| | Aff-F | **0.854** | 0.852 | 0.847 | 0.840 | 0.827 | 0.679 | 0.724 | 0.675 | 0.831 | 0.845 | 0.841 | 0.844 | 0.439 | 0.742 | 0.626 | 0.738 | 0.629 |

# E. Additional model analysis

## E.1. Effect of the Number of Periods and Convolution Kernels

Modeling multiple periods is beneficial for capturing diverse temporal dependencies in time series data. Figure 5a reports the model performance under different numbers of periods $N$. It can be observed that considering the top three dominant periods yields strong performance, whereas incorporating additional periods beyond this point does not yield further performance improvements. Besides, we further investigate the impact of the number of convolution kernels $R$. As shown in figure 5b, setting $R = 3$ is sufficient to achieve strong performance, while further increasing $R$ yields no significant performance gains and may instead reduce computational efficiency.

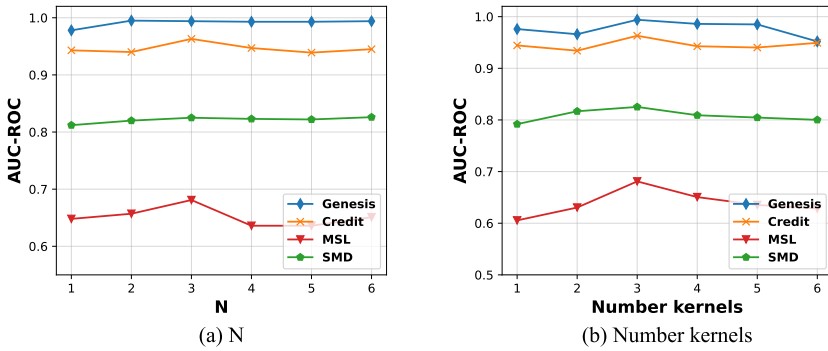

(a) N        (b) Number kernels

*Figure 5.* Visualization comparisons of different numbers of periods and convolution kernels.

## E.2. Additional ablation experiments

Such as Tabel 11, we conducted ablation experiments on all datasets, and the results demonstrated the effectiveness of each module design.

*Table 11.* Ablation study on different design choices.

| Variations | | GECCO | Genesis | CaIlt2 | Credit | MSL | SMAP | PSM | SMD | Avg |
|---|---|---|---|---|---|---|---|---|---|---|
| **Channel Variations** | CI | 0.946 | 0.886 | 0.826 | 0.913 | 0.658 | 0.542 | 0.688 | 0.792 | 0.781 |
| | CD | 0.983 | 0.938 | 0.814 | 0.948 | 0.664 | 0.537 | 0.676 | 0.812 | 0.796 |
| | CC | 0.980 | 0.885 | 0.846 | 0.921 | 0.590 | 0.583 | 0.704 | 0.809 | 0.790 |
| | w/o $\mathbf{X}_M$ | 0.809 | 0.727 | 0.684 | 0.552 | 0.595 | 0.491 | 0.647 | 0.740 | 0.656 |
| | w/o $\mathbf{X}_A$ | 0.983 | 0.957 | 0.830 | 0.948 | 0.646 | 0.584 | 0.706 | 0.802 | 0.807 |
| | random assignment | 0.924 | 0.886 | 0.721 | 0.905 | 0.625 | 0.514 | 0.671 | 0.811 | 0.757 |
| **Interaction Variations** | w/o extraction | 0.980 | 0.992 | 0.829 | 0.909 | 0.646 | 0.587 | 0.707 | 0.823 | 0.809 |
| | w/o gate layer | 0.970 | 0.985 | 0.827 | 0.908 | 0.627 | 0.581 | 0.698 | 0.820 | 0.802 |
| | add technique | 0.962 | 0.958 | 0.815 | 0.905 | 0.616 | 0.564 | 0.662 | 0.804 | 0.786 |
| | $\mathbf{X}_A$ influences $\mathbf{X}_M$ | 0.981 | 0.993 | 0.828 | 0.947 | 0.636 | 0.572 | 0.702 | 0.812 | 0.809 |
| **$\mathbf{X}_A$ modeled same as $\mathbf{X}_M$** | | 0.985 | 0.942 | 0.844 | 0.951 | 0.652 | 0.573 | 0.677 | 0.821 | 0.806 |
| **Masking Variations** | point masker | 0.973 | 0.978 | 0.835 | 0.943 | 0.647 | 0.575 | 0.671 | 0.812 | 0.804 |
| | segment masker | 0.983 | 0.981 | 0.816 | 0.954 | 0.648 | 0.570 | 0.670 | 0.820 | 0.805 |
| | period masker | 0.982 | 0.983 | 0.831 | 0.942 | 0.622 | 0.569 | 0.696 | 0.819 | 0.805 |
| | w/o masker | 0.966 | 0.963 | 0.792 | 0.916 | 0.627 | 0.551 | 0.6613 | 0.791 | 0.783 |
| **TeamWork (ours)** | | **0.992** | **0.994** | **0.850** | **0.963** | **0.681** | **0.592** | **0.720** | **0.825** | **0.827** |

# F. Visualization case studies

To enable intuitive performance comparison, we conduct a comparative visualization of anomaly scores between TeamWork and various recent SOTAs, including CrossAD, CATCH, and ModernTCN. As shown in Figure 6, the blue line is the input time series, and the pink area indicates the locations of the anomalies. The first row shows the original series, the second to fourth rows display anomaly scores from the baselines, and the last row is our model. Teamwork exhibits the most distinguishable anomaly scores in detecting both point and subsequence anomalies.

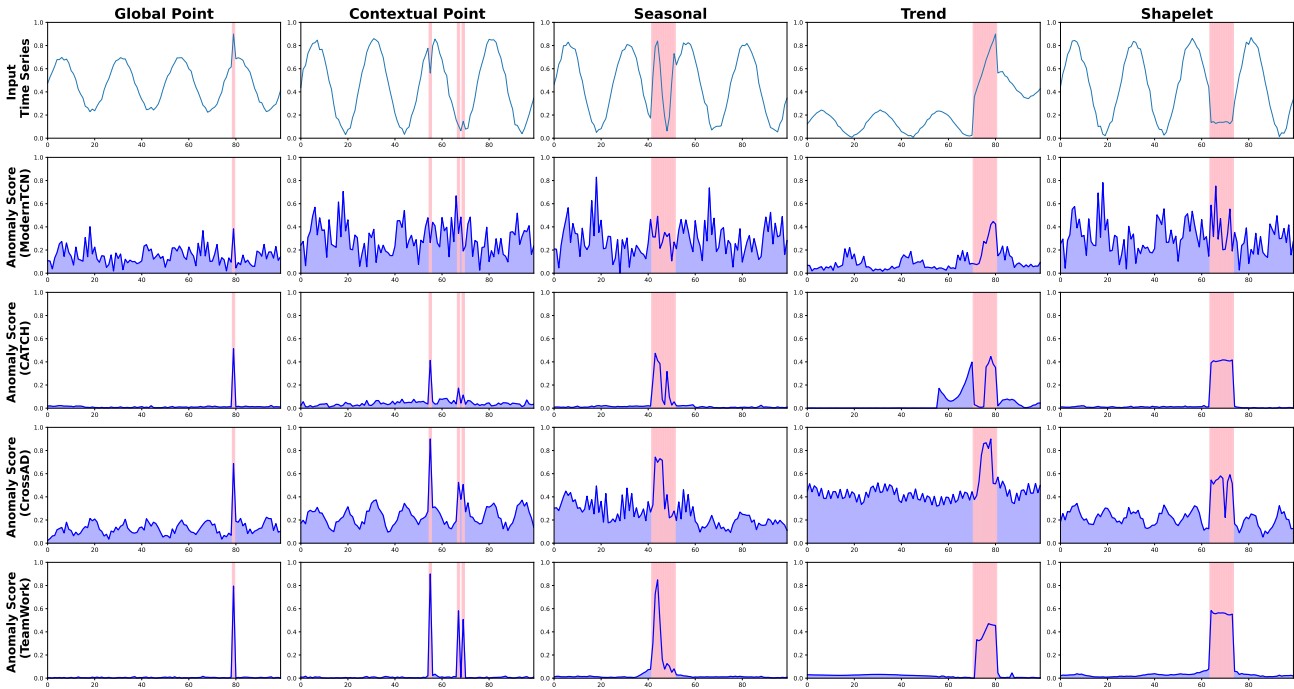

*Figure 6.* Visualization comparisons of anomaly scores between TeamWork and recent SOTAs across five anomaly patterns.

