# OpenReview forum: "TeamWork: Multivariate Time Series Anomaly Detection via Asymmetric Role-aware Channel Modeling"
_ICML.cc/2026/Conference — ICML 2026 regular_

### Official Review · Reviewer_Naot · 2026-02-15

**Soundness:** 3
**Presentation:** 2
**Significance:** 2
**Originality:** 2
**Overall Recommendation:** 3
**Confidence:** 4

**Summary:**

This paper addresses the core challenges in multivariate time series anomaly detection (MTSAD), namely the complexity of channel relationship modeling, the overlooked relative importance of variables, and the uncaptured divergent manifestations of anomalies in multi-periodic time series, and proposes an asymmetric role-aware channel modeling framework termed TeamWork. Based on the maximum entropy principle, this framework partitions variables into dominant variables and auxiliary variables, devises a period-aware masked modeling mechanism to adapt to the anomaly characteristics of time series across different periods, and fuses the information of dominant variables and auxiliary variables through a role-aware gated interaction module. When distinguishing between dominant and auxiliary variables, since the Entropy Reduction Index (ERI) is intractable to compute directly, it is converted into the calculation of mutual information, with the Kraskov estimator adopted to derive its approximation. After the processed of relative ratio normalization, the normalized ERI values are compared with a preset threshold to complete the discrimination of dominant and auxiliary variables. In the period-aware masked modeling, the dominant periods of time series are extracted via the Fast Fourier Transform (FFT), followed by the tailored design of a point masker, a segment masker and a period masker for short, medium and long periods respectively. Combined with a multi-conv encoder, the framework captures temporal dependencies at different granularities, which effectively adapts to the diverse manifestations of anomalies in multi-periodic systems. Finally, a role-aware gated interaction module is designed to supplement the information of dominant variables with that of auxiliary variables, thereby facilitating the reconstruction of dominant variables. The reconstruction error is directly employed as the anomaly score, which is subsequently compared with a predefined anomaly threshold to classify each time point as normal or anomalous. Extensive experiments on eight real-world datasets demonstrate that TeamWork outperforms the current state-of-the-art methods for MTSAD.

**Compliance With Llm Reviewing Policy:**

Affirmed.

**Final Justification:**

My recommendation is Weak Reject. The authors’ rebuttal has not fully addressed my key concerns.The model proposed in this paper is essentially merely an engineering-level combination of existing components, lacking fundamental theoretical innovation.In addition, the work introduces an extra threshold to distinguish between dominant variables and auxiliary variables, as well as a new hyperparameter $m$. I am deeply concerned about this practice of introducing additional thresholds and hyperparameters, as it will significantly increase the difficulty of real-world application.

**Key Questions For Authors:**

1. This paper partitions the variables of multivariate time series (MTS) into dominant variables and auxiliary variables based on the Entropy Reduction Index (ERI). However, the sequence patterns of each variable in MTS vary significantly. How to prove, from both theoretical derivation and experimental verification perspectives, that the variable role partitioning based on ERI can truly reflect the practical roles of each variable in system evolution, and that this partitioning method possesses rationality and universality in MTS data with different feature distributions and periodic characteristics?
2. In this paper, the ERI threshold for variable partitioning and the masking ratio are both determined to their optimal values via parameter sensitivity experiments on the test set. Nevertheless, in practical multivariate time series anomaly detection (MTSAD) tasks, only training sets dominated by normal data are generally available, with no prior information from the test set. How to reasonably set and adaptively adjust the aforementioned hyperparameters in pure unsupervised training set scenarios to ensure the effectiveness of the model in practical deployment?
3. This paper uses reconstruction error as the anomaly score. What is the specific calculation method for the anomaly score of a single time step? Is it the arithmetic mean of the reconstruction errors of all variables at the given time step? Please clearly illustrate how the anomaly score of a single time step is calculated. In addition, general unsupervised reconstruction methods determine whether a time step is anomalous by comparing the anomaly score with a predefined threshold, yet this paper does not explicitly elaborate on the selection method of the anomaly decision threshold. What strategy is adopted to determine this threshold? And is there any corresponding experimental verification for the effectiveness of this threshold selection strategy?

**Limitations:**

Explicit limitations/future work are not articulated in the main paper body. It is recommended that the authors conduct a discussion on the limitations of each module and appropriately clarify how to optimize against these limitations of each module in future research.

**Strengths And Weaknesses:**

This paper demonstrates sound rigor in terms of theoretical support, method selection and model evaluation. Specifically, the design of the core methods is underpinned by explicit theoretical foundations and detailed technical specifications. To address the intractability of direct calculation of the Entropy Reduction Index (ERI), the paper proves the equivalence between ERI and mutual information through rigorous mathematical derivation, and further adopts the Kraskov estimator to approximate the multivariate mutual information. Clear mathematical definitions and operational procedures are provided for modules including the period-aware masker and the role-aware gated interaction module: for instance, the dominant periods of time series are extracted via the Fast Fourier Transform (FFT); differentiated mathematical formulas of masks are tailored for time series across different periods; and sigmoid activation is applied in the gated interaction to realize selective information fusion. All technical implementation details are specific and feasible.
On the theoretical front, the paper defines dominant and auxiliary variables based on the maximum entropy principle, and completes the formula derivation of ERI as well as the mathematical modeling of the masking strategy, laying a solid theoretical foundation for the design of the TeamWork framework. From the experimental perspective, the experimental design is scientific and rational: the study selects 8 real-world datasets covering domains such as industrial system monitoring, financial fraud detection and spacecraft anomaly detection, and conducts comparisons with 16 state-of-the-art baseline methods, which cover models adopting the channel-independent (CI), channel-dependent (CD) and channel-clustering (CC) channel modeling strategies. A comprehensive evaluation is performed using more than ten label-based and score-based metrics including the Affiliated-F1 score (Aff-F), the Area under the Receiver Operating Characteristics Curve (A-R) and the F1-score. In addition, auxiliary experiments such as ablation studies, parameter sensitivity analysis and anomaly score visualization are designed to verify the effectiveness of the TeamWork framework and all its sub-modules.
The writing and presentation of this paper comply with the norms of machine learning academic papers, featuring clear organization and easy comprehensibility. The narrative is straightforward, and the visualizations provide excellent auxiliary support for understanding. In the related work section, the field of multivariate time series anomaly detection (MTSAD) is systematically categorized from two levels: the first level classifies MTSAD methods into forecasting-based, contrastive-based and reconstruction-based paradigms, and the second level divides channel modeling strategies into CI, CD and CC types. This classification thoroughly analyzes the limitations of existing methods. Meanwhile, the paper explicitly elaborates on the core differences between the proposed channel role-aware (CR) strategy and traditional channel modeling strategies, as well as between the period-aware masking mechanism and the existing fixed masking strategies, laying a clear literature foundation for the innovations of this work. The authors have also made an anonymous code repository publicly available and promised to release the complete code upon paper acceptance, which provides sufficient information for expert readers in the field to reproduce the experimental results and meets the reproducibility requirements of high-quality academic papers.
The research direction explored in this paper bears significant theoretical and practical implications, yet the complexity of the model design to a certain extent limits its application value and domain influence. This paper addresses the core challenges in the MTSAD field, thus possessing prominent practical value. The proposed asymmetric role-aware channel modeling paradigm breaks the inherent cognition of "treating all variables equally" in traditional channel modeling, reveals the law of "dominant-auxiliary" role differentiation of variables in MTSAD, and enriches the academic understanding of channel relationship modeling in this field. Additionally, the design of the period-aware masking mechanism provides a novel methodological paradigm for solving the anomaly detection problem of multi-periodic time series data.
Despite the notable contributions of this research, the model architecture is overly complex with an obvious issue of module stacking: it integrates multiple sophisticated modules including ERI-based variable decoupling, FFT-based period extraction, the tailored design of point, segment and period maskers, the multi-conv encoder, and the role-aware gated interaction module. Each module is designed with elaborate details, such as distinct mathematical definitions for different maskers and a two-layer information extraction and fusion mechanism in the gated interaction. This not only increases the overall complexity of the model but also raises the threshold for its deployment in real-world scenarios. Furthermore, in addition to the anomaly detection threshold, an extra threshold is introduced to distinguish dominant variables from auxiliary variables, and the masking ratio also needs to be preset, which may significantly increase the complexity of hyperparameter tuning. The operations of variable differentiation and period-aware masking entail certain risks, and multiple modules are strongly coupled with each other—if one module malfunctions, it is highly likely to cause the failure of the entire system.
In terms of originality, this paper proposes a new perspective for MTSAD channel modeling, designs a novel strategy for multi-period anomaly capture, and creatively combines these new designs with classic ideas in the field. By integrating the asymmetric role-aware channel modeling, the multi-period-aware masking strategy and the role-aware gated interaction for MTSAD, the paper improves the reconstruction quality of time series and enhances the discriminability between normal and abnormal data. The work exhibits distinct differences from existing studies and possesses novelty.
To sum up, the research ideas and methods of this paper provide a novel perspective of treating different variables in a differentiated manner for the MTSAD field, and are supported by rigorous theoretical demonstrations and comprehensive experiments. However, the overall model is overly complex due to the stacking of multiple sophisticated modules including ERI-based variable decoupling, FFT-based period extraction, the design of point, segment and period maskers, the multi-conv encoder, and the role-aware gated interaction module. Moreover, given the significant heterogeneity of features in multivariate time series data, different features may exhibit completely distinct patterns. Thus, the processes of variable differentiation, period calculation and mask application based on dominant periods are accompanied by certain risks—erroneous variable differentiation or improper period analysis is likely to lead to the failure of the entire system. It is recommended that the authors add relevant discussions on the limitations of the feature differentiation module and the period-aware masking module, as well as corresponding solution strategies.

---

> ### Author Rebuttal · Authors · 2026-03-31
>
> We sincerely thank Reviewer Naot for the recognition of our work and for providing constructive comments. Below, we address your concerns and questions individually:
>
> # W1: The model architecture is complex.
> Our design addresses core challenges in multivariate anomaly detection, and each component serves a specific purpose:
> 1. Variable heterogeneity: ERI-based decoupling separates dominant and auxiliary variables, and dominant-guided extraction reduces redundancy in the auxiliary branch.
> 2. Multi-period anomalies: We extract dominant periods from dominant variables and assign tailored maskers (point/segment/period) to short/mid/long periods, aligning masking with scale-specific anomaly patterns instead of using a one-size-fits-all strategy.
> 3. Multi-scale fusion: A multi-conv encoder captures temporal patterns at different scales, and role-aware gated interaction fuses dominant and auxiliary representations.
>
> Thus, the architecture is principled and necessary to handle the diverse and hierarchical nature of multivariate time series.
>
> # W2 & Q2: How to adjust the hyperparameters in pure unsupervised training set scenarios?
> The sensitivity analysis (Figure 4 in the paper) demonstrates that TeamWork is robust across a wide range of values for the mask ratio $m$ and dominance threshold $\delta^*$, not that optimal values require test-set tuning.
>
> Moreover, if $m$ is too small, the reconstruction task becomes too easy; if too large, the task becomes hard. If $\delta^*$ is too small, redundant variables may enter the dominant set; if too large, useful signals may be excluded.
>
> In practice, setting both parameters around 0.5 typically yields strong performance without fine-tuning. We will clarify this in the revised manuscript.
>
> # W3: If one of the modules fails, it is likely to lead to the failure of the entire system.
> We clarify that it does not lead to a catastrophic failure of the entire system in practice. As explicitly demonstrated in the ablation study (Table 3 in the paper), even when removing the variable partitioning module (Channel Variations) or the period-aware masking module (Masking Variations), the performance understandably degrades. But the degraded variants remain competitive with the baselines.
>
> # Q1: How about the rationality and universality of this partitioning method?
> Theoretical level:
>
> ERI is derived from entropy $\mathcal{H}(\mathbf{X}{-{x_{i,:}}})$ and conditional entropy $\mathcal{H}(\mathbf{X}{-{x_{i,:}}} \mid x_{i,:})$ (Eq.(1) in the paper), which quantifies how much a variable reduces uncertainty about rest of the system, reflecting the representational ability of this variable for multivariate system. As shown in Appendices B.1 and B.2, ERI does not rely on a specific distributional assumption of a particular dataset. Moreover, normalizing ERI by its maximum value removes scale inconsistency across variables. These properties support the rationality and broad applicability of our method across MTS data with various feature distributions and periodic characteristics.
>
> Experimental level:
> - TeamWork performs strongly on eight real-world datasets with diverse domains and periods. (**Rebuttal Table G https://anonymous.4open.science/r/TeamWork-3F84/Rebuttal/G.png; Table 1 in the paper**).
> - As shown in **Rebuttal Table H** (https://anonymous.4open.science/r/TeamWork-3F84/Rebuttal/H.png), (1) removing the dominant variables, (2) replacing ERI with random partition, and (3) using similarity-based clustering all degrade performance.
>
> These results provide support for the rationality and applicability of TeamWork's partition design across diverse datasets.
>
> # Q3: Anomaly score and anomaly decision threshold.
> For each time step $t$, the anomaly score is computed by aggregating the reconstruction errors of all variables:
> $$s_t=\frac{1}{C}\sum_{c=1}^{C}(x_{t,c}-\hat{x}_{t,c})^2.$$
>
> Following the standard benchmark[1], we report both score-based metrics (e.g., A-R, threshold-free) and label-based metrics, which require thresholding. For the latter, we select the best threshold from a predefined range ([0.1,0.5,1,2,3,5,10,15,20,25]%) for fair comparison. Teamwork performs strongly under both settings.
>
> [1] TAB: Unified Benchmarking of Time Series Anomaly Detection Methods, *Proc. VLDB Endowment*, 2025.
>
> # Limitation.
> - Limitation of role assignment: TeamWork currently adopts a static role perception for variables. In highly complex dynamic systems, a variable that is dominant in one stage may become auxiliary in another stage.
> - Limitation of the period-aware masking module: The current design mainly focuses on global periodicity and may be less effective when the series has no clear periodic structure.
> - Future work: We will investigate a dynamic role-aware mechanism that allows variable roles to evolve.
> - Future work: We will explore combining the current design with time-frequency analysis tools, such as wavelet transforms, to better capture local temporal variations.

---

> > ### Author Rebuttal · Reviewer_Naot · 2026-04-01
> >
> > 1. None of the three core modules of this work is particularly novel, they essentially amount to an engineering-level combination of existing components, lacking fundamental theoretical innovation and exhibiting an apparent issue of module stacking.
> >
> > 2. In addition to the anomaly decision threshold for anomaly scoring, the authors introduce an extra threshold to distinguish dominant variables and auxiliary variables, along with a new hyperparameter $m$. As the authors themselves acknowledge: if $m$ is too small, the reconstruction task becomes too easy; if too large, the task becomes hard. If $\delta^*$ is too small, redundant variables may enter the dominant set; if too large, useful signals may be excluded. I hold serious concerns about this practice of introducing additional thresholds and hyperparameters, as it substantially increases the difficulty of real-world application.

---

> > > ### Author Response · Authors · 2026-04-01
> > >
> > > Thank you for your response! We would like to address your concerns as follows:
> > >
> > > ### 1. On Model Novelty
> > >
> > > TeamWork introduces a novel perspective on **inter-channel relationships** in multivariate time series by proposing the first information-theoretic decoupling mechanism. Unlike existing methods limited to channel-independent (CI) or channel-dependent (CD) paradigms and overlooking **the relative importance** of different variables, TeamWork adopts a **role-aware modeling** strategy with two principled steps:
> > >
> > > 1. **Dominant variable selection**: First uses ERI to partition the multivariate system into system-driving complementary (**dominant**) and complementary variables (**auxiliary**).
> > > 2. **Role-aware modeling**: Applies tailored modeling strategies for different roles of variables.
> > >
> > > Overall, this design is principled and necessary, not mere module stacking. As noted by Reviewer Xprt ("**well-founded motivation**", "**sufficient theoretical elaboration**"), Reviewer aZs8 ("**new insights**"), and Reviewer 1UPr ("**novel**").
> > >
> > > ### 2. On Hyperparameter Burden
> > >
> > > Our sensitivity analysis (Fig. 4) shows that TeamWork is **highly stable**: using default values of 0.5 for both $m$ and $\delta^*$ consistently yields near-optimal performance **across diverse datasets**. In practice, these parameters can be **fixed at 0.5 without tuning**, imposing no difficulty of real-world application.
> > > The "too large/too small" examples in our earlier rebuttal were illustrative to explain why extreme values (e.g., 0.1 or 0.9) are avoided, not suggestions for actual tuning ranges.
> > >
> > > Thank you again for your meticulous review.

---

### Official Review · Reviewer_1UPr · 2026-02-28

**Soundness:** 2
**Presentation:** 2
**Significance:** 2
**Originality:** 2
**Overall Recommendation:** 3
**Confidence:** 4

**Summary:**

This paper proposes TeamWork to tackle the difficulty of multivariate time‑series anomaly detection by explicitly modeling variable relationships and temporal dependencies. It first separates variables into dominant and auxiliary roles based on how much each contributes to reducing uncertainty, and proposed a role‑aware gated interaction module to fuse these two groups. To handle point and subsequence anomalies that appear across multiple periodic patterns, TeamWork introduces a period‑aware masked modeling mechanism that employs a suite of specialized masking strategies spanning short to long periods, enabling thorough temporal‑dependency learning. Experiments on real‑world datasets demonstrate the promising results from TeamWork.

**Compliance With Llm Reviewing Policy:**

Affirmed.

**Final Justification:**

I increase my review score. However, I still have concerns about its practicality on real‑world time series.

**Key Questions For Authors:**

- Section 4.1: How to define the threshold for $\delta$ in  equation 2?
- Section 4.2: The three types of masks, equations 5, 6, 7, are independent. Is is possible to have them support each other?

**Limitations:**

The paper has a section on impact statement.

**Strengths And Weaknesses:**

The paper tackles two important practical problems in time series anomaly detection: modeling the multi‑time‑series correlations and the diverse manifestations of anomalies in multi‑periodic systems.

The authors propose the role‑aware channel modeling to address the first challenge. While the technique appears novel, I have strong concerns about its practicality on real‑world time series. The main issue lies in the criteria for separating dominant and auxiliary variables. The entropy‑reduction index (ERI) is based on a single‑variable test, which ignores the “group contribution” to the system dynamics. A simple example illustrates the problem: if a dominant variable has a near duplicate in the multivariate series, the method described in the paper will miss that variable. I encourage the authors to significantly revise this approach to make it applicable in real‑world settings.

Regarding the period‑aware masking model, the technique makes sense but I do not consider it to be highly original. There are also concerns about the practicality of the proposed methods. In Section 4.2, FFT is applied to each time series to discover the periods \(P_{1}, P_{2}, \dots\) for each series. However, different time series (or groups) exhibit different periods. How can unified periods for the multivariate series be identified? The authors need to explain this clearly.

---

> ### Author Rebuttal · Authors · 2026-03-31
>
> We sincerely thank Reviewer 1UPr for recognizing our work and for the constructive feedback. We address the concerns as follows.
>
> # W1: Group contribution.
> We clarify that ERI is not designed to search all possible variable groups for the largest joint contribution. Instead, we aim to identify the variable that most effectively reduces the uncertainty of the multivariate system, thereby characterizing each variable's explanatory power for the overall system dynamics.
>
> In the case where a dominant variable has a near-duplicate, our goal is still met: we seek representative variables, not redundant copies. In addition, a thresholding strategy based on the maximum ERI value helps retain variables with similarly strong dominant effects. This aligns with the dominant/auxiliary decoupling objective, where additional complementary signals are in the auxiliary branch.
>
> Importantly, this case does not contradict our formulation. Consider two variables $x_{i,:}$ and $x_{j,:}$. Based on the equivalence between ERI and mutual information (Appendix B.1), and the chain rule of mutual information, we have $$I\left({x_{i,:}, x_{j,:}}; \mathbf{X}_ {-(x_{i,:}, x_{j,:})}\right)=I\left(x_{i,:}; \mathbf{X}_ {-(x_{i,:}, x_{j,:})}\right) + I\left(x_{j,:}; \mathbf{X}_ {-(x_{i,:}, x_{j,:})} \mid x_{i,:}\right).$$
>
> If $x_{i,:}$ and $x_{j,:}$ are nearly identical, the conditional term satisfies $$I\left(x_{j,:}; \mathbf{X}_ {-(x_{i,:}, x_{j,:})} \mid x_{i,:}\right) \to 0.$$ Which implies $$I\left({x_{i,:}, x_{j,:}}; \mathbf{X}_ {-(x_{i,:}, x_{j,:})}\right)\approx I\left(x_{i,:}; \mathbf{X}_ {-(x_{i,:}, x_{j,:})}\right).$$
>
> Thus, the joint contribution of the near-duplicate pair is approximately equal to that of $x_{i,:}$, resolving the concern.
>
> # W2: About the highly originality of the period-aware masking model.
> Our innovation lies in linking multi-period anomaly manifestation to period-specific masking assignment. Existing time series masking methods are typically period-agnostic (random point masking or fixed patch masking).
>
> Anomalies manifest differently across periods: point anomalies are more pronounced in short-period views but likely to be smoothed in long-period representations, while subsequence anomalies are clearer at longer periods-especially when their duration exceeds short-period lengths. TeamWork addresses this by tailoring masking to the underlying periodicity.
>
> Ablation results (Table 3 in the paper) show that using any single masker (point, segment, or period) underperforms the full model, confirming the benefit of period-aware joint masking. Visualizations (Figure 3 in the paper) further validate TeamWork's ability to accurately detect both point and subsequence anomalies.
>
> # W3: How can unified periods for the multivariate series be identified?
> In the framework, period calculation is applied only to the dominant variables $X_M$. For each variable in $X_M$, we apply the FFT along the temporal dimension, transform it into the frequency domain, and calculate the amplitude of the resulting frequency components. Then, average the amplitudes across the variable dimension to obtain a unified global amplitude spectrum. The top-$N$ frequencies with the highest amplitudes are selected and converted into unified period lengths.
>
> # Q1: How to define the threshold $\delta$ for in equation 2?
> We do not use the absolute ERI value $\Delta_i^{\mathcal H}$ directly as the threshold, as they may vary across datasets due to differing scales and distributions. Instead, we normalize each ERI by the maximum ERI value to measure the relative dominance of each variable.
>
> This threshold design offers the following benefits: 1. eliminating the scale inconsistency of original ERI values across variables, while preserving the absolute magnitude of entropy reduction. 2. Normalization by the maximum ERI is order-preserving, maintains the original ranking of variables' dominance, i.e., $\Delta_i^{\mathcal H} \le \Delta_j^{\mathcal H} \iff \delta_i \le \delta_j$. Moreover, dominant variables only guide role-aware modeling rather than to directly determine anomaly labels, reducing sensitivity to threshold choice.
>
> A sensitivity study in [0.1, 0.9] (Section 5.3, Fig. 4(a)) shows that a moderate value (0.5) strikes the best balance and yields optimal performance.
>
> # Q2: Is it possible to have three types of masks that support each other?
> We added an cross-mask interaction variant. As shown in **Rebuttal Table F** (https://anonymous.4open.science/r/TeamWork-3F84/Rebuttal/F.png), it does not yield clear gains, likely because masking patterns across mismatched periods introduces interference.
>
> Our original design applying dedicated, independent masking strategies per branch preserves their distinct inductive biases. Meanwhile, these branches are not isolated, their encoded representations are aligned and aggregated into the final dominant-variable representation (Section 4.2, Eq.(9)).

---

> > ### Author Rebuttal · Reviewer_1UPr · 2026-04-06
> >
> > I increased my score

---

> > > ### Author Response · Authors · 2026-04-06
> > >
> > > Thank you for carefully reading our previous response and for providing your valuable feedback. We are committed to addressing every concern raised regarding the manuscript. Every question, doubt, and suggestion is of great significance to us, and we sincerely appreciate the opportunity to engage in this constructive dialogue.
> > >
> > > We are happy that you raised the original score, and deeply value your engagement and dedication as a reviewer for ICML 2026. If our revision meets your expectations, would you consider giving us a score marginally above accept?😉

---

### Official Review · Reviewer_Xprt · 2026-03-09

**Soundness:** 3
**Presentation:** 3
**Significance:** 3
**Originality:** 3
**Overall Recommendation:** 5
**Confidence:** 4

**Summary:**

This paper proposes a channel role-aware mechanism that explicitly classifies variables into dominant variables and auxiliary variables based on the maximum entropy principle. Furthermore, to capture the diverse manifestations of anomalies in multi-periodic systems, the paper introduces a period-aware masked modeling mechanism. Extensive experiments on multiple benchmark datasets demonstrate the superior performance of TeamWork in multivariate time series anomaly detection tasks.

**Compliance With Llm Reviewing Policy:**

Affirmed.

**Final Justification:**

The authors' rebuttal has addressed my concerns, and I will keep my initial score.

**Key Questions For Authors:**

1. Would processing auxiliary variables with more complex modeling approaches be conducive to performance improvement?
2. What would be the effect if $H_A$ also exerts an influence on $H_M$ in the Role-aware Gated Interaction Module?
3. The authors need to explain whether important variables may be misclassified as unimportant ones.

**Limitations:**

yes

**Strengths And Weaknesses:**

Strengths:
1. The paper proposes an asymmetric role-aware channel modeling framework with a well-founded motivation is introduced.
2. Sufficient theoretical elaboration. The proofs presented in this work lay a solid mathematical foundation for decoupling multivariate time series into dominant variables and auxiliary variables.
3. Well-structured experimental design. Extensive experiments are conducted on multiple datasets, against a robust set of 16 baseline models and with 16 evaluation metrics adopted.
4. The paper is clearly written and logically organized, with a coherent flow that makes the motivation well understood.

Weaknesses:
1. Auxiliary variables are processed with a lightweight MLP, and it is necessary to discuss the impact of more complex modeling methods on representation learning.
2. The authors partition multivariate variables into dominant variables and auxiliary variables. If all variables in a dataset are equally important (i.e., all are dominant variables), there is a high risk of misclassifying some equally important variables as unimportant ones, which would thereby limit the modeling capability of the framework.

---

> ### Author Rebuttal · Authors · 2026-03-31
>
> We sincerely thank Reviewer Xprt for recognizing our work and for the constructive feedback. We address the concerns as follows.
>
> # W1 & Q1: Auxiliary variables use more complex modeling methods.
> As mentioned in Table 3 of the original paper, we have also considered using the more complex period-aware masked modeling to process the auxiliary variables. As suggested, we further include the results on all datasets under this setting (see **Rebuttal Table D** https://anonymous.4open.science/r/TeamWork-3F84/Rebuttal/D.png). The results show that applying more complex modeling to the auxiliary variables does not bring clear performance improvement. This is likely because auxiliary variables contain relatively limited information, and overly fine-grained modeling may instead introduce noise and hinder learning.
>
> # W2 & Q3: Misclassification cases.
> Thank you for this valuable comment. We would like to clarify that the purpose of the proposed ERI-based partition is not to assume that some variables must be unimportant, but to distinguish variables according to their relative explanatory power with respect to the overall multivariate system. ERI is designed to identify the variables that most effectively reduce the uncertainty of the remaining system. In the case you mentioned, where many or even all variables are similarly important, our design is still applicable. Since we use a thresholding strategy based on normalization by the maximum ERI value, variables with similarly strong dominant effects can be retained together in the dominant set, rather than forcing only a few variables to be selected.  Therefore, the framework does not require the dominant set to be small.
>
> We will clarify this point more explicitly in the revised paper.
>
> # Q2: $H_A$ also exerts an influence on $H_M$.
> Following your advice, we further implemented a variant in which $\mathbf{H}_A$ also influences $\mathbf{H}_M$. As shown in **Rebuttal Table E** (https://anonymous.4open.science/r/TeamWork-3F84/Rebuttal/E.png), this bidirectional interaction does not bring clear performance improvement over the original design. A possible reason is that $\mathbf{H}_M$ already represents the dominant variables, introducing additional guidance from $\mathbf{H}_A$ may instead bring redundant information into the dominant branch. This is also consistent with our original design motivation: the role-aware interaction is intended to use the representations of dominant variables to guide the learning of auxiliary features, thereby enhancing the complementary information in auxiliary variables that is relevant to the dominant ones.

---

> > ### Author Rebuttal · Reviewer_Xprt · 2026-04-03
> >
> > Thank the authors for their detailed rebuttal. Our concerns have been addressed, and we maintain our score.

---

> > > ### Author Response · Authors · 2026-04-03
> > >
> > > Many thanks to Reviewer Xprt for your prompt response and your positive support. We would like to once again express our sincere gratitude for taking the time to review our paper and for providing insightful comments.

---

### Official Review · Reviewer_aZs8 · 2026-03-13

**Soundness:** 2
**Presentation:** 3
**Significance:** 2
**Originality:** 3
**Overall Recommendation:** 4
**Confidence:** 4

**Summary:**

The paper proposed a novel framework for multivariate time series anomaly detection by first classifying the variables, based on the estimated entropy reduction, into dominant variables and auxiliary variables, and then, based on such classification, incorporating the role-aware grade interaction module. Additionally, to capture period-specific anomalies, three different period maskers are applied. The experiments on seven real-world datasets show the capability of the proposed model.

**Compliance With Llm Reviewing Policy:**

Affirmed.

**Final Justification:**

My concerns have been adequately addressed.

**Key Questions For Authors:**

1. The parameter sensitivity analysis shows that the model is not sensitive to either the dominant-variable selection threshold $\delta^*$ or the time series mask ratio $m$. Using a threshold of 0.1, which selects many dominant variables, gives similar performance to using a threshold of 0.9, which selects only a few dominant variables, and masking 90% of the periods gives similar performance to masking 10% of the periods. This is suggested by the nearly horizontal lines with little variation in Figure 4. This behavior shows that the accuracy of masking and selection is not important.
2. It is hard to tell whether the improvement is significant over the seven real-world datasets.
3. According to the results reported in the CrossAD paper, CrossAD achieves better performance than the proposed algorithm on SMAP (A-R), PSM (Aff-F), GECCO (V-ROC, V-PR), and SMD (V-ROC, V-PR). This might be due to different hyperparameter settings?
4. Could you explain what the dominant-guided extraction block is used for? Does the asymmetry come from this block? In the ablation study, what is the model framework without the extraction block? In this case, is $H_A$ used in Eq. 12 instead of $H_A^*$?
5. It would be more complete to include an ablation study on all seven datasets, rather than a subset of them.

Others:

Another paper submitted to ICML 2026 is mistakenly included here as supplementary material.

**Limitations:**

No. The authors have not adequately discussed the limitations of their work, although a discussion of potential negative societal impact may not be necessary.

**Strengths And Weaknesses:**

*Soundness*

1. There is no theoretical analysis, and I think this might be a common phenomenon in this area? The experimental results on seven real-world datasets, compared with multiple baselines, show that the proposed model has strong performance.
2. The ablation study shows that each module in the framework contributes to the strong performance of the model.
3. The parameter sensitivity analysis shows that the model is not sensitive to either the dominant-variable selection threshold $\delta^*$ or the time series mask ratio $m$.

However, there are some concerns about:

1. whether the increase in performance compared with existing work is significant,
2. and the fact that the model is not sensitive to $\delta^*$ and $m$ raises a new question about whether the selection and masker are needed, as a threshold of 0.1 has the same performance as a threshold of 0.9.
3. The evaluation on CrossAD also raises some questions.


Please refer to the Key Question section for details.

*Presentation*

The paper is well written, and the plots are clear. The logic of the paper is also clear.

*Significance*

Anomaly detection in time series is an interesting topic. Separating variables based on entropy, as well as using a period-specific masker, is also practical in real-world applications. However, the prerequisite for these two modules is based on estimation independent of the proposed framework. More specifically, it is hard to measure the uncertainty and accuracy of the KE estimator needed for selecting variables, as well as for choosing periods based on the Fast Fourier Transform.

*Originality*

The work provides new insights through role-awareness and period-awareness.

---

> ### Author Rebuttal · Authors · 2026-03-31
>
> We appreciate your recognition of the TeamWork framework and its strong empirical performance across diverse datasets. Below, we address your concerns and questions individually:
>
> # W1 & Q2: The increase in performance compared with existing work.
> As shown in **Rebuttal Table A** (https://anonymous.4open.science/r/TeamWork-3F84/Rebuttal/A.png), report the average results of all methods in Table 1 of the paper. Compared with the best baseline, TeamWork improves the performance by 4.7% in Aff-F1 and 4.0% in A-R. For reference, the gap between the best and the second-best baseline is only 0.9% and 4.1% on these two metrics. Additionally, TeamWork achieves competitive or superior results in 16 total evaluation metrics (Table 2 in the paper), further demonstrating its effectiveness.
>
> # W2 & Q1: The parameter sensitivity analysis shows that the accuracy of masking and selection is not important.
> We add ablations to verify the necessity of the masking and variable selection: (1) w selection_CI/CD: remove variable selection; (2) Random: replace ERI-based selector with random partition; (3) w masker: remove maskers.
>
> As shown in **Rebuttal Table B** (https://anonymous.4open.science/r/TeamWork-3F84/Rebuttal/B.png):
> - Removing variable selection (CI/CD) degrades performance.
> - Random partition results in a drop, as it cannot accurately identify the dominant variables.
> - Removing masker hurts performance, because the learning task would become overly simplistic, failing to learn meaningful representations.
>
> We clarify that similar performance at 0.1 and 0.9 does not mean these modules are unnecessary, it indicates robustness over a broad range.
>
> # W3 & Q3: The evaluation on CrossAD.
> To ensure a fair comparison, all baselines, including CrossAD, were re-evaluated under the unified benchmark[1]. Therefore, some deviations are expected, and we believe such a unified evaluation is more objective and reliable than directly comparing results from different papers under different settings.
>
> [1] TAB: Unified Benchmarking of Time Series Anomaly Detection Methods, *Proc. VLDB Endowment*, 2025.
>
> # W4: Issues with KE and period estimation reliability.
> We clarify that TeamWork does not rely on exact entropy values, but on the relative ranking induced by ERI to select dominant variables. The KE estimator is also nonparametric and does not require distributional assumptions. Moreover, auxiliary variables are not discarded; they still contribute through the Role-aware Gated Interaction Module. Ablation study (Table 3 in paper) supports the effectiveness of this design.
>
> For period selection, in the absence of prior knowledge, existing methods[2] indicated that the period length can be derived through techniques such as the FFT. Moreover, TeamWork uses the top-N periods rather than relying on a single one, so even if one is imperfectly estimated, others still useful. Figure 5 (in paper) shows the effectiveness of the design.
>
> [2] LightGTS: A Lightweight General Time Series Forecasting Model, ICML, 2025.
>
> # Q4: Some questions about the dominant-guided extraction block.
> (1) The dominant-guided extraction block uses dominant variable representations to enhance complementary information in the auxiliary branch.
> (2) The asymmetry does not come from this block alone. ERI-based variable decoupling first assigns asymmetric roles, and this block further reinforces that design.
> (3) Yes. Without this block, $\mathbf{H}_A$ is directly used in Eq.(12) instead of $\mathbf{H}_A^*$. In the paper Table 3, w extraction means removing this block, which reduces the average performance from 0.927 to 0.908, confirming the effectiveness of the module.
>
> # Q5: Ablation study on all datasets.
> Following the suggestion, we add ablation studies for all datasets. As shown in **Rebuttal Table C** (https://anonymous.4open.science/r/TeamWork-3F84/Rebuttal/C.png), we can draw the same conclusions as in the original ablation analysis, which fully validates the effectiveness of the design of each module.
>
> # Other: Another paper submitted to ICML 2026 is mistakenly included here as supplementary material.
> According to the official ICML policy, papers with overlapping authorship must be treated as prior work, and anonymized PDFs must be included in the supplementary material. We therefore included the other paper accordingly.
>
> # Limitations.
> We add the following limitations and future work:
> - Limitations: TeamWork currently adopts a static role perception for variables. In highly complex dynamic systems, a variable that is dominant in one stage may become auxiliary in another stage. The current design mainly focuses on global periodicity and may be less effective when the series has no clear periodic structure.
> - Future work: We will investigate a dynamic role-aware mechanism that allows variable roles to evolve. We will explore combining the current design with time-frequency analysis tools, such as wavelet transforms, to better capture local temporal variations.

---

> > ### Author Rebuttal · Reviewer_aZs8 · 2026-04-03
> >
> > Thank you to the authors for the detailed response and the additional experimental results
> >
> > Most of my concerns have been addressed by the rebuttal and the new results. I have two remaining questions, one of which was also raised by Reviewer 1UPr.
> >
> > 1. Could the authors provide a more detailed explanation of why "the fact that the model is not sensitive to $\delta^*$ and $m$ raises a new question about whether the selection and masker are needed, as a threshold of 0.1 has the same performance as a threshold of 0.9"? At the moment, the rebuttal mainly states the conclusion that the model is robust to these parameters. However, I am still unclear about the mechanism behind this. Why is performance largely insensitive to such a wide range of thresholds, while removing these modules altogether still leads to a noticeable drop in performance? Since 0.1 and 0.9 are quite far apart, does this suggest that there may be a threshold region, for example around 0.05, beyond which performance improves and then remains stable?
> >
> > 2. Regarding the response to the question, "How can unified periods for the multivariate series be identified?", does this imply that the model is better suited to multivariate time series in which the dominant variables share a consistent period, since the method relies on a shared period structure?

---

> > > ### Author Response · Authors · 2026-04-03
> > >
> > > We sincerely thank Reviewer aZs8 for the insightful comments and continued engagement. Our responses are provided below.
> > >
> > > ### 1. Detailed explanation of parameters.
> > > (1) **About the mask ratio.**
> > > The key point is that the sensitivity analysis changes the **masking strength**, not whether the **period-aware masking mechanism** is used. Thus, the observed stability reflects the robustness of our masking mechanism, not that maskers are unnecessary.
> > >
> > > Our method does not rely on a single masker. Instead, it employs three specialized maskers to capture diverse anomaly manifestations in multi-periodic systems. Although changing the mask ratio alters the reconstruction difficulty, the multi-period structure still remains. It allows the model to learn effective self-supervision across different temporal resolutions. This is why performance varies only mildly over a broad range. However, removing three maskers entirely (**w/o masker**) leads to a clear drop, demonstrating that maskers are indeed effective.
> > >
> > > Following the reviewer's suggestion, we additionally test m=0.05. Performance drops further at this extreme value, showing that the model is not insensitive to all settings, but rather has a relatively broad yet bounded threshold region. When masking is too weak, the model can rely too heavily on visible local points and fail to learn deeper representation. By contrast, **m=0.5** gives consistently strong results across datasets, so we recommend it as a practical default without further tuning.
> > >
> > > **Table 1.** Results under mask ratio is 0.05.
> > > | Method  |  GECCO | Genesis | CaILt2 | Credit | MSL | SMAP | PSM | SMD |
> > > | :--------- | :------: | :---------: | :----------: | :-----------: |  :---------: | :----------: | :-----------: | :-----------: |
> > > | m = 0.05 |0.970|0.971|0.810|0.933|0.642|0.560|0.691|0.803|
> > > |**TeamWork (ours)**|0.992| 0.994 | 0.850 | 0.963 | 0.681 | 0.592 | 0.720 | 0.825 |
> > >
> > > (2) **About the threshold.**
> > > In our framework, variables are not "selected or discarded," but assigned to different roles. All variables remain involved in anomaly detection. The threshold $\delta^* $ identifies the most representative variables that drive the system dynamics, while the remaining variables are still modeled in the auxiliary branch and adaptively fused through the **role-aware gated interaction module**. Therefore, performance remains relatively stable over a broad range of $\delta^* $.
> > >
> > > By contrast, **w/o selection** completely removes the dominant-auxiliary decoupling and models all variables uniformly, which leads to a clear performance drop. Following the reviewer's suggestion, we also tested $\delta^*=0.05$, and performance also declines. This suggests that extremely low thresholds weaken the role of decoupling and introduce more redundancy.
> > >
> > > In summary, relative stability over a broad parameter range reflects the **robustness** of the design, while the noticeable drops after removing the modules show that the modules themselves are indeed **necessary**. These two observations are not contradictory.
> > >
> > > **Table 2.** Results under $\delta^*$ is 0.05.
> > > | Method  |  GECCO | Genesis | CaILt2 | Credit | MSL | SMAP | PSM | SMD |
> > > | :--------- | :------: | :---------: | :----------: | :-----------: |  :---------: | :----------: | :-----------: | :-----------: |
> > > | $\delta^*$ = 0.05 |0.976|0.968|0.828|0.938|0.661|0.565|0.691|0.790 |
> > > |**TeamWork (ours)**|0.992| 0.994 | 0.850 | 0.963 | 0.681 | 0.592 | 0.720 | 0.825 |
> > >
> > >
> > > ### 2. About the period.
> > > For each variate in $X_M$, we apply the FFT along the temporal dimension to transform it into the frequency domain, and calculate the amplitude of the resulting frequency components. Then, the amplitudes are **averaged across the variate dimension** to obtain a global amplitude spectrum. The top-$N$ frequencies with the highest amplitudes are selected and converted into corresponding period lengths.
> > >
> > > It is worth noting that this period extraction procedure does not strictly require all variates to share identical periodicities. Instead, the averaging operation is designed to capture the dominant temporal scales representative of the entire system. Moreover, TeamWork **selects multiple periods** rather than a single one, which further means it is not restricted to a shared period structure.
> > >
> > > We will incorporate these clarifications and additional results into the final version.

---

### Decision · Program_Chairs · 2026-04-30

**Decision:**

Accept (regular)

**Comment:**

This submission studies multivariate time-series anomaly detection and proposes a method that combines role-aware channel modeling with period-aware temporal modeling. The reviewers generally agreed that the paper addresses a relevant problem, is clearly presented, and is supported by a reasonably extensive empirical evaluation. Several reviewers found the perspective on distinguishing variable roles interesting and potentially useful, and the paper’s experimental results were considered competitive.

The main discussion centered on the degree of conceptual novelty and on the complexity and practicality of the overall design. One reviewer remained unconvinced that the work rises beyond a careful integration of existing ideas, while others were more positive and felt that the rebuttal adequately addressed their concerns. Overall, although this is a borderline case and the conceptual contribution is somewhat modest, the paper meets the bar as a technically solid and relevant contribution likely to be of interest to the community.